# The homeodomain transcriptional regulator DVE-1 directs a program for synapse elimination during circuit remodeling

Kellianne D. Alexander [1,2], Shankar Ramachandran[1], Kasturi Biswas [1,2], Christopher M. Lambert[1], Julia Russell[1], Devyn B. Oliver [1,2], William Armstrong[1], Monika Rettler[1], Samuel Liu [1,2], Maria Doitsidou [3], Claire Bénard[1,4], Amy K. Walker[5] & Michael M. Francis [1,2] ✉

The elimination of synapses during circuit remodeling is critical for brain maturation; however, the molecular mechanisms directing synapse elimination and its timing remain elusive. We show that the transcriptional regulator DVE-1, which shares homology with special AT-rich sequence-binding (SATB) family members previously implicated in human neurodevelopmental disorders, directs the elimination of juvenile synaptic inputs onto remodeling *C. elegans* GABAergic neurons. Juvenile acetylcholine receptor clusters and apposing presynaptic sites are eliminated during the maturation of wild-type GABAergic neurons but persist into adulthood in *dve-1* mutants, producing heightened motor connectivity. DVE-1 localization to GABAergic nuclei is required for synapse elimination, consistent with DVE-1 regulation of transcription. Pathway analysis of putative DVE-1 target genes, proteasome inhibitor, and genetic experiments implicate the ubiquitin-proteasome system in synapse elimination. Together, our findings define a previously unappreciated role for a SATB family member in directing synapse elimination during circuit remodeling, likely through transcriptional regulation of protein degradation processes.

The mature human brain is composed of billions of neurons that are organized into functional circuits based on stereotyped patterns of synaptic connections that optimize circuit performance. Mature circuit connectivity is choreographed through a remarkable period of developmental circuit rewiring that is broadly conserved across species[1–4]. During this rewiring or remodeling phase, the mature circuitry is established through a tightly controlled balance: on the one hand, degenerative processes promote the elimination of juvenile synapses, while on the other hand, maintenance or growth processes support the stabilization or formation of new connections. A

combination of cell-intrinsic and extrinsic factors shape the progression of these events. For instance, activity-dependent microglial engulfment and elimination of synaptic material shapes connectivity of the retinogeniculate system in mice[5–8], while cell-intrinsic genetic programs such as the circadian clock genes Clock or Bmal1 influence GABAergic maturation and plasticity-related changes in the neocortex[9]. Whereas molecular mechanisms supporting axon guidance and synapse formation have received considerable attention, our understanding of neuron-intrinsic molecular mechanisms controlling synapse elimination remains more limited. In particular, it is

[1]Department of Neurobiology, University of Massachusetts Chan Medical School, Worcester, MA, USA. [2]Program in Neuroscience, University of Massachusetts Chan Medical School, Worcester, MA, USA. [3]Centre for Discovery Brain Sciences, University of Edinburgh, Edinburgh, Scotland. [4]Department of Biological Sciences, Université du Québec à Montréal, Quebec, Canada. [5]Program in Molecular Medicine, University of Massachusetts Chan Medical School, Worcester, MA, USA. ✉e-mail: Michael.Francis@umassmed.edu

unclear how neuron-intrinsic synapse elimination processes are engaged in developing neural circuits. Improved mechanistic knowledge of these processes offers potential for important advances in our grasp of brain development. This knowledge may also inform the pathology underlying numerous neurodevelopmental diseases associated with altered connectivity and neurodegenerative diseases where synapse loss is a hallmark feature. Indeed, recent work has suggested intriguing parallels between the elimination of synapses during development and neurodegenerative processes during disease[10–13].

The nematode *Caenorhabditis elegans* offers significant assets for addressing mechanistic questions about developmental neural circuit remodeling, particularly synapse elimination. *C. elegans* progress through a highly stereotyped period of nervous system remodeling that establishes the neural connections characteristic of mature animals. 80 of the 302 neurons composing the adult nervous system, including 52 motor neurons, are born post-embryonically and integrated into pre-existing juvenile circuits following the first larval (L1) stage of development[14,15]. The incorporation of these post-embryonic born motor neurons is accomplished through a remarkable reorganization of circuit connectivity. One of the most striking aspects of this reorganization is the remodeling of synaptic connections in the GABAergic dorsal D-class (DD) motor neurons (Fig. 1a)[16,17]. Immediately after hatch, juvenile cholinergic synaptic inputs onto GABAergic DD neurons are located dorsally, and juvenile DD synaptic outputs onto muscles are located ventrally. During remodeling, the juvenile dorsal cholinergic synaptic inputs onto DD neurons are eliminated, and new synaptic inputs from post-embryonic-born presynaptic cholinergic neurons are established ventrally. In parallel, ventral DD GABAergic synaptic terminals are relocated dorsally, forming new GABAergic synaptic contacts onto dorsal muscles[18–22]. Though we now have a growing appreciation of the cellular processes that direct the post-embryonic redistribution of DD GABAergic outputs onto dorsal muscles, we have a limited understanding of how cholinergic inputs onto DD neurons are remodeled. Prior work suggested a mechanism for antagonizing the remodeling of cholinergic inputs onto DD neurons through temporally controlled expression of the Ig domain family member OIG-1[23,24]; however, the mechanisms that promote remodeling of these inputs, in particular their elimination, have remained uncharacterized.

We report the identification of a mechanism for neuron-intrinsic transcriptional control of synapse elimination during remodeling of the *C. elegans* motor circuit. From a forward genetic screen to isolate mutants whose juvenile postsynaptic sites remain present on mature GABAergic DD neurons, we obtained a mutation in the homeodomain transcriptional regulator *dve-1* that shares homology with mammalian special AT-rich sequence-binding (SATB) family members, which are implicated in human neurodevelopmental disorders such as SATB2-associated syndrome[25,26]. We show that DVE-1 acts cell autonomously in GABAergic DD neurons to promote the removal of juvenile cholinergic synaptic inputs. Juvenile synaptic inputs are maintained into adulthood in *dve-1* mutants, leading to an accumulation of presynaptic cholinergic material and accompanying effects on circuit function and movement. We further show that precocious synapse elimination in *oig-1* mutants is reversed by mutation of *dve-1*, suggesting that DVE-1 promotes pro-degenerative processes that are antagonized by OIG-1. Our results reveal a neuron-intrinsic mechanism for the regulation of neurodevelopmental synapse elimination through the actions of a conserved homeodomain transcriptional regulator.

## Results

### Distinct mechanisms direct developmental remodeling of presynaptic terminals versus postsynaptic sites in GABAergic neurons

Previous work by our lab showed that clusters of postsynaptic ionotropic acetylcholine receptors (iAChR) denote postsynaptic sites on DD neurons[23,27,28]. These postsynaptic sites undergo dorsoventral remodeling during the transition between the 1st and 2nd larval stages of *C. elegans* development (L1/L2 transition)[23,24]. During this period, dorsal postsynaptic sites on DD neurons are removed, and new ventral postsynaptic sites are formed as indicated by the appearance of new ventral iAChR clusters (Fig. 1a). The dorsoventral remodeling of cholinergic postsynaptic sites in DD neurons occurs coincidently with the ventrodorsal rearrangement of GABAergic presynaptic terminals (labeled by the synaptic vesicle marker mCherry::RAB-3) (Fig. 1a; S1.1A, B). Notably, we found that mutations in several genes previously implicated in the remodeling of GABAergic presynaptic terminals had no appreciable effect on the remodeling of cholinergic postsynaptic sites in DD neurons (Fig. S1.1C, Table 1). For example, juvenile cholinergic postsynaptic sites are properly removed from the dorsal processes of DD neurons in *ced-3*/caspase mutants (Fig. S1.1C), while juvenile RAB-3 clusters persist in the ventral nerve cord until much later in development (through the L4 stage)[29]. Indeed, lingering synaptic vesicle clusters in the ventral nerve cord of *ced-3* mutants are interleaved with newly formed ventral iAChR clusters at L4 stage (Fig. S1.1C), demonstrating that the formation of new ventral postsynaptic sites during remodeling also occurs independently of *ced-3*. Mutations in several genes important for neurotransmitter release and calcium signaling also do not appreciably alter the remodeling of postsynaptic sites in DD neurons, though we noted clear delays in the remodeling of DD presynaptic terminals as found previously[19,21] (Table 2). Of the genes we tested, only mutation of the RyR/*unc-68* gene produced a modest delay in the remodeling of postsynaptic sites, suggesting calcium release from intracellular stores contributes (Table 2). Taken together, our findings demonstrate that mechanisms for remodeling postsynaptic sites in DD neurons are distinct from those previously implicated in the remodeling of DD GABAergic presynaptic terminals.

### Identification of DVE-1 as a transcriptional regulator of synapse elimination

Motivated by these findings, we performed a forward mutagenesis screen to identify previously undefined mechanisms controlling the removal of juvenile postsynaptic sites in DD neurons (Fig. S1.2A, B). From this screen we isolated a recessive mutant, *uf171*, where juvenile postsynaptic sites, indicated by dorsally positioned iAChR clusters, are not properly eliminated during remodeling. Dorsal postsynaptic sites are normally eliminated before the L2 stage (22 h after hatch) in wild type but remain visible through the late L4 stage (>40 h after hatch) in *uf171* mutants (Fig. 1b). Whole genome sequence analysis of *uf171* mutants revealed a point mutation that produces a proline to serine (P/S) substitution in the gene encoding the homeodomain protein DVE-1 (Fig. 1c). Expression of the wild-type *dve-1* cDNA in *dve-1(uf171)* mutants using either the native promoter region or a GABA-specific promoter restored the normal elimination of juvenile postsynaptic sites (Fig. 1d, e; Fig. S1.2C), while *dve-1* overexpression in wild-type animals did not produce appreciable changes in removal (Fig. 1d; Fig. S1.2C, D). A similar failure in the elimination of juvenile postsynaptic sites is also evident using another available *dve-1* mutant, *dve-1(tm4803)*, that harbors a small insertion/deletion mutation (Fig. 1c, e; Fig. S1.2C, D). The P/S substitution encoded by *dve-1(uf171)* affects a highly conserved proline residue predicted to lie within a loop between helices I and II of the first homeodomain of DVE-1. *dve-1(tm4803)* deletes a portion of predicted helix III in the same homeodomain and a splice site, leading to a 65 bp insertion (Fig. 1c)[30]. As *dve-1* null mutants are embryonic lethal[31], both mutations are predicted to be hypomorphic. Our identification of *dve-1* and further analysis demonstrate a cell-autonomous requirement for *dve-1* in DD GABAergic neurons for the neurodevelopmental elimination of juvenile postsynaptic receptors during remodeling. *dve-1* was of particular interest because it encodes a homeodomain transcriptional regulator sharing homology with

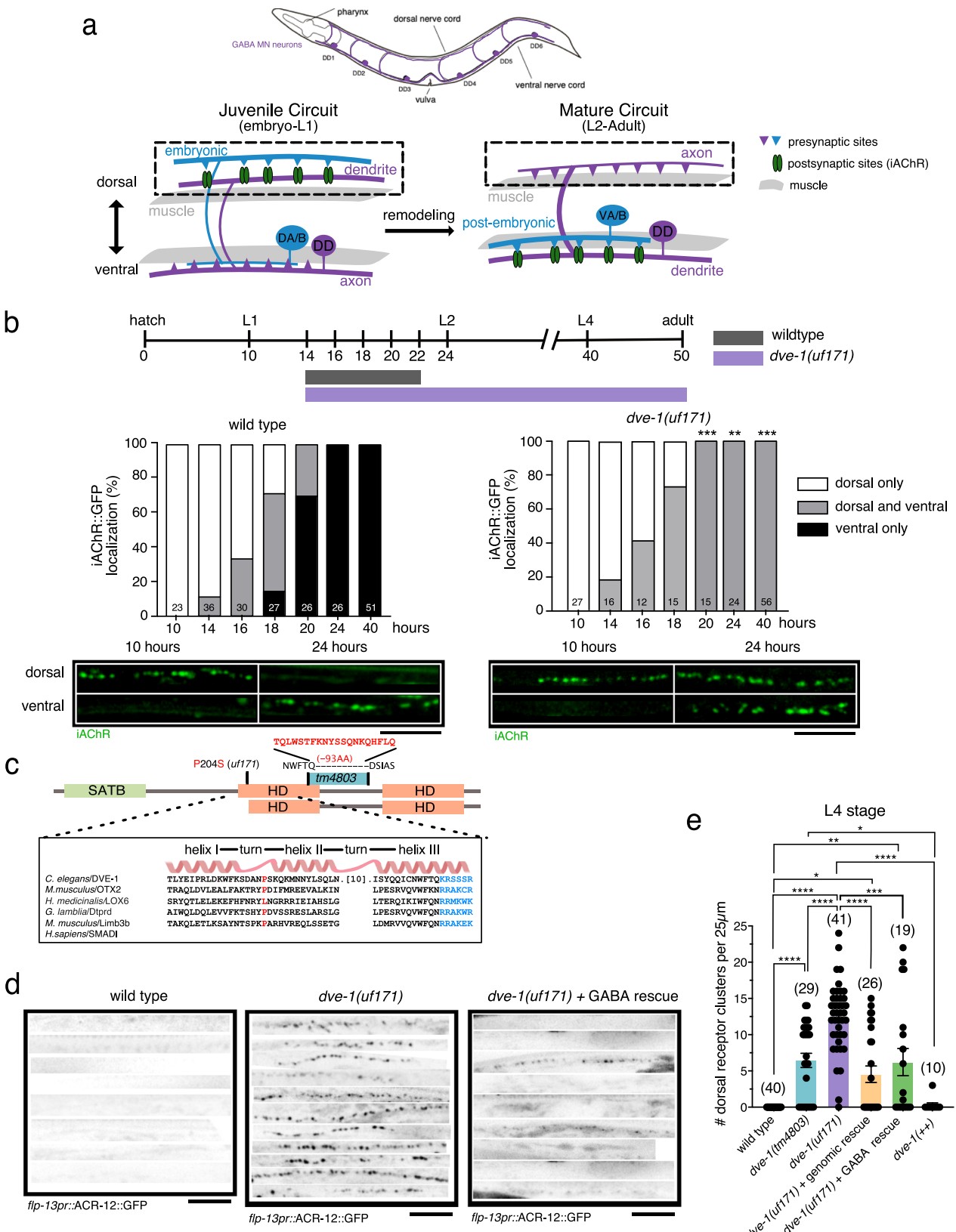

mammalian SATB transcription factors that have roles in vertebrate neurodevelopment[25,26]. In *C. elegans*, the sole previously characterized function for *dve-1* is in the regulation of a mitochondrial stress response[31].

To better define the requirement for DVE-1 in synapse elimination we next used the AID (auxin inducible degron) system for

spatiotemporally controlled DVE-1 degradation (Fig. 2a–f). In this system, a plant F-box protein, TIR1, mediates auxin-dependent degradation of AID-tagged proteins[32,33]. We used an engineered *dve-1::AID::wrmScarlet* allele in combination with pan-neuronally expressed *TIR1::BFP::AID* to control DVE-1 degradation (Fig. 2a). Consistent with prior *dve-1* expression analysis[34], under control conditions

**Fig. 1 | Mutation of the homeodomain transcription factor *dve-1* disrupts the removal of postsynaptic sites in GABAergic motor neurons. a** Top, schematic of *C. elegans* indicating DD GABAergic motor neurons (purple). Bottom, schematic depicting motor circuit before (left) and after (right) remodeling. DD motor neurons (purple), cholinergic motor neurons (blue). **b** Top, timeline of remodeling, approximate timing of transitions between larval stages and adulthood are indicated. Bars indicate the duration of DD synaptic remodeling for wild-type (gray) and *dve-1* mutants (purple). Elimination of dorsal cord iAChR clusters is completed by 22 h after hatch for wild-type whereas dorsal iAChR clusters persist through adulthood in *dve-1* mutants. Middle, quantification of iAChR remodeling in DD neurons of wild-type (left) and *dve-1* mutants (right). *X*-axis time from hatch in hours. Animals are binned according to the distribution of iAChR puncta as dorsal only (white), ventral only (black), or dorsal and ventral (gray). Bottom, representative images of dorsal and ventral ACR-12::GFP (iAChRs, green) clusters for wild-type (left) and *dve-1* mutants (right) at the times indicated. Two-tailed Fischer's

exact test with Bonferroni correction. Scale bar, 5 μm. **c** Domain structure of DVE-1. SATB-like domains and homeodomains (HD) are indicated. Site of substitution produced by *uf171* missense mutation (red) and region of *tm4803* deletion mutation (blue) and insertion (red) are indicated. Box, predicted protein structure (AlphaFold) and sequence alignment for HD1 (NCBI Conserved Domains). **d** Fluorescent confocal images of synaptic iAChR clusters in GABAergic DD processes of the dorsal nerve cord at L4 stage. GABA rescue refers to specific expression of wild-type *dve-1* cDNA using the *unc-47* promoter. In this and subsequent figures, iAChR refers to ACR-12::GFP unless otherwise indicated. Images on each line are from different animals. Scale, 5 μm. **e** Quantification of the number of iAChR clusters per 25 μm of the L4 stage dorsal nerve cord for the genotypes indicated. Each dot represents a single animal and n for each genotype is indicated by numbers in parentheses. Bars indicate mean ± SEM. ****$p < 0.0001$, ***$p < 0.001$, **$p < 0.01$, *$p < 0.05$, one-way ANOVA with Tukey's multiple comparisons test.

**Table 1 | Mutations that delay GABAergic presynaptic remodeling do not affect cholinergic postsynaptic remodeling in DD motor neurons**

| AChR (*flp-13pr*::ACR-12::GFP) | | | | | |
|---|---|---|---|---|---|
| Genotype (L4 stage) | Percent remodeled (ventral only) (n) | Percent not remodeled (dorsal only and both sides) (n) | Total (n) | Fischer's Exact *P*-value (two-tailed) | Bonferroni Correction (significant?) $\alpha < 0.003$ |
| Wild type | 100 (100) | 0 (3) | 103 | | |
| *ced-3(ok2734)* | 100 (21) | 0 (0) | 21 | 1 | No |
| *ced-3(n717)* | 100 (23) | 0 (0) | 23 | 1 | No |
| *cdk-5(ok626)* | 100 (20) | 0 (0) | 20 | 1 | No |
| *unc-8(e49)* | 100 (20) | 0 (0) | 20 | 1 | No |
| *unc-8(e15)* | 100 (20) | 0 (0) | 20 | 1 | No |
| SV (*flp-13pr*::mCherry::RAB-3) | | | | | |
| Genotype (L4 stage) | Percent remodeled (ventral only) (n) | Percent not remodeled (dorsal only and both sides) (n) | Total (n) | Fischer's Exact *P*-value | Bonferroni Correction (significant?) $\alpha < 0.01$ |
| Wild type | 100 (40) | 0 (0) | 40 | | |
| *ced-3(ok2734)* | 56 (6) | 44 (5) | 11 | 0.0003 | Yes |
| *ced-3(n717)* | 57 (13) | 43 (10) | 23 | 8.95E-06 | Yes |
| *cdk-5(ok626)* | 24 (5) | 76 (16) | 21 | 1E-10 | Yes |
| *unc-8(e49)* | 56 (9) | 44 (7) | 16 | 5E-05 | Yes |
| *unc-8(e15)* | 62 (13) | 38 (8) | 21 | 0.0001 | Yes |

*dve-1::AID::wrmScarlet* is expressed in the nuclei of both intestinal cells and ventral cord neurons–solely in the DD neuron nuclei in the ventral nerve cord at L1 stage, and in both VD and DD nuclei at L4 stage (Fig. 2c, d). The remodeling of postsynaptic iAChR clusters proceeds normally in these animals in the absence of auxin (Fig. 2e, f). Continuous auxin treatment for ~50 h from hatch strikingly decreased both *TIR1::BFP::AID* and *DVE-1::AID::wrmScarlet* levels in neurons. Intestinal *DVE-1::AID::wrmScarlet* was not significantly affected (Fig. 2c, d), demonstrating that degradation was specific to TIR1 expressing cells. Continuous treatment with auxin for either 50 or 24 h after hatch severely disrupted synapse elimination in these animals (Fig. 2e, f). In contrast, auxin treatment after the completion of remodeling (beginning ~24 h after hatch) did not impact the dorsoventral distribution of receptor clusters (Fig. 2e). Our findings indicate that neuronal DVE-1 is required prior to and/or during the remodeling period for synapse elimination to proceed but DVE-1 is not required later in development to maintain the mature organization of postsynaptic receptors in the circuit.

The postsynaptic scaffold protein LEV-10 is associated with cholinergic postsynaptic sites in body wall muscles and GABAergic neurons[35,36]. Using a previously characterized *lev-10* allele that enables cell-specific endogenous labeling with split-GFP[36], we investigated the removal of LEV-10 during remodeling (Fig. 2g). At L1 stage, LEV-10 is primarily associated with dorsal DD processes in wild type and is then redistributed to ventral processes during remodeling. Similar to

juvenile iAChR clusters, dorsal LEV-10 scaffolds in GABAergic neurons of *dve-1* mutants are not properly eliminated during remodeling, demonstrating that DVE-1 coordinates the removal of both juvenile postsynaptic receptors and associated proteins. Interestingly, mutation of *dve-1* does not significantly affect ventral postsynaptic sites on DD neurons that are formed during remodeling (Fig. 2h; Fig. S2.1A). Likewise, mutation of *dve-1* has little effect on the density of DD dendritic spines that are formed at the end of remodeling (Fig. S2.1B)[27,37,38]. Thus, DVE-1 governs the elimination of juvenile postsynaptic sites during remodeling without affecting the formation or maturation of new postsynaptic sites. Notably, the remodeling of DD GABAergic presynaptic terminals also occurs normally in *dve-1* mutants (Fig. 2H, Fig. S2.1C), further indicating that distinct neuron-intrinsic programs direct remodeling of the pre- and postsynaptic domains of GABAergic DD neurons.

### Lingering iAChRs in *dve-1* mutants are organized into structural synapses

To test if lingering juvenile postsynaptic sites in the dorsal nerve cord of *dve-1* mutants are organized into structurally intact synapses, we first asked whether these iAChR clusters are localized at the cell surface. We found that lingering dorsal iAChR clusters in *dve-1* mutants could be labeled by in vivo injection of antibodies to an engineered extracellular HA epitope[38,39], suggesting localization at the cell surface (Fig. 3a). We also found that most of the iAChR clusters retained in

## Table 2 | Effects of synaptic activity and calcium signaling on synaptic remodeling

**AChR (*flp-13pr*::ACR-12::GFP)**

| Genotype (24 h post-hatch) | Percent remodeled (ventral only) (n) | Percent not remodeled (dorsal only and both sides) (n) | Total (n) | Fischer's Exact P-value (two-tailed) | Bonferroni Correction (significant?) α < 0.0055 |
|---|---|---|---|---|---|
| Wild type | 96 (64) | 4 (3) | 67 | | |
| *unc-2(e55)* | 89 (16) | 11 (2) | 18 | 0.29 | No |
| *cca-1(ad1650)* | 100 (17) | 0 (0) | 17 | 1 | No |
| *itr-1(sa73)* | 100 (18) | 0 (0) | 18 | 1 | No |
| *unc-68(e540)* | 50 (10) | 50 (10) | 20 | 9.02E-06 | Yes |
| *unc-13(e51)* | 100 (15) | 0 (0) | 15 | 1 | No |
| *unc-43(e408)* | 95 (19) | 5 (1) | 20 | 1 | No |
| *unc-18(e234)* | 100 (16) | 0 (0) | 16 | 1 | No |
| *unc-17(e113)* | 76 (13) | 24 (4) | 17 | 0.03 | No |

| Genotype (L4 stage) | Percent remodeled (ventral only) (n) | Percent not remodeled (dorsal only and both sides) (n) | Total (n) | Fischer's Exact P-value (two-tailed) | Bonferroni Correction (significant?) α < 0.003 |
|---|---|---|---|---|---|
| Wild type | 100 (93) | 0 (0) | 93 | | |
| *unc-2(e55)* | 100 (20) | 0 (0) | 20 | 1 | No |
| *unc-2(zf35)* | 88 (15) | 12 (2) | 17 | 0.15 | No |
| *cca-1(ad1650)* | 100 (17) | 0 (0) | 17 | 1 | No |
| *itr-1(sa73)* | 90 (18) | 10 (2) | 20 | 0.18 | No |
| *unc-68(e540)* | 90 (19) | 10 (2) | 21 | 0.2 | No |
| *unc-43(e408)* | 95 (18) | 5 (1) | 19 | 0.17 | No |
| *unc-43(e498)* | 100 (15) | 0 (0) | 15 | 1 | No |
| *unc-18(e234)* | 100 (16) | 0 (0) | 16 | 1 | No |
| *unc-31(e928)* | 100 (20) | 0 (0) | 20 | 1 | No |
| *unc-17(e113)* | 100 (15) | 0 (0) | 15 | 1 | No |

**AChR (*flp-13pr*::ACR-12::GFP)**

| Genotype (24 h post-hatch) | Percent remodeled (ventral only) (n) | Percent not remodeled (dorsal only and both sides) (n) | Total (n) | Fischer's Exact P-value (two-tailed) | Bonferroni Correction (significant?) α < 0.025 |
|---|---|---|---|---|---|
| Wild type | 97 (28) | 3 (1) | 29 | | |
| *unc-18(e234)* | 44 (7) | 56 (9) | 16 | 0.0001 | Yes |
| *unc-17(e113)* | 33 (2) | 67 (4) | 6 | 0.0014 | Yes |

dorsal GABAergic DD processes of *dve-1* mutants are in close apposition with synaptic vesicle assemblies and active zones in cholinergic axons of the dorsal nerve cord, suggesting incorporation into structural synapses (Fig. 3b–d).

During remodeling, cholinergic DA/B connections with DD neurons in the dorsal nerve cord are removed, and new DA/B connections are established with post-embryonic born ventrally directed GABAergic D-class (VD) motor neurons (Fig. 3e). To investigate how cholinergic presynaptic terminals may be affected by mutation of *dve-1*, we expressed the photoconvertible synaptic vesicle marker Dendra2::RAB-3 in cholinergic DA/B neurons (Fig. 3f–j; Fig. S3.1A–G). We first examined the distribution of Dendra2::RAB-3 in the wild-type dorsal nerve cord immediately prior to the onset of DD remodeling (~14 h after hatch). Prior to photoconversion, clusters of green Dendra2::RAB-3 fluorescence were distributed along the length of cholinergic axons in the dorsal nerve cord (Figure S3.1A). Brief exposure to 405 nm light produced immediate and irreversible photoconversion of Dendra2::RAB-3 from green to red fluorescence (Figs. 3g–j and S3.1B–G). In wild type, Dendra2::RAB-3 clusters that had been photoconverted to red fluorescence prior to the onset of remodeling were strikingly reduced following remodeling (10 h later, 55 ± 9% reduction) and were replaced by new synaptic vesicle clusters (green fluorescence) (Fig. 3g, h; Fig. S3.1B–D). In contrast, wild-type Dendra2::RAB-3 clusters photoconverted after the completion of remodeling (at ~24 h after hatch) remained largely stable over the subsequent 10 h (Fig. 3i, j; Fig. S3.1E–G). New green RAB-3 clusters also became visible during this

time frame (24–34 h after hatch), indicating a parallel addition of new vesicular material (Fig. 3i, j; Fig. S3.1F, G). Thus, synaptic vesicle clusters in wild-type DA/B axons are largely removed and replaced during the 10-h period of remodeling but are more stable over a 10-h time window immediately following completion of remodeling, offering intriguing evidence for developmental stage-specific regulation of cholinergic synaptic vesicle stability.

We noted a striking change in the stability of synaptic vesicle material in cholinergic axons of *dve-1* mutants during remodeling. Most dorsal cholinergic Dendra2::RAB-3 clusters photoconverted prior to the onset of synaptic remodeling were preserved throughout remodeling in *dve-1* mutants (Fig. 3g, h; Fig. S3.1B–D), indicating enhanced stability of cholinergic terminals presynaptic to DD neurons. The addition of new synaptic vesicles during this time frame (14–24 h after hatch) was not appreciably affected by mutation of *dve-1*, as indicated by similar increases in green Dendra2::RAB-3 fluorescence across wild-type and *dve-1* mutant cholinergic axons (Fig. 3g, h; Fig. S3.1C, D). RAB-3 clusters that were photoconverted after the completion of remodeling (~24 h after hatch) remained detectable 10 hours later in *dve-1* mutants, also similar to wild type (Fig. 3i, j; Fig. S3.1E–G). Green Dendra2::RAB-3 fluorescence in dorsal axons increased by roughly 2-fold in *dve-1* mutants compared to wild type at 34 h after hatch, suggesting enhanced addition or stabilization of new synaptic vesicles at *dve-1* mutant cholinergic axon terminals over the 10 hrs following remodeling (Fig. 3i, j; Fig. S3.1F, G). Consistent with this observation, we noted that the intensity of

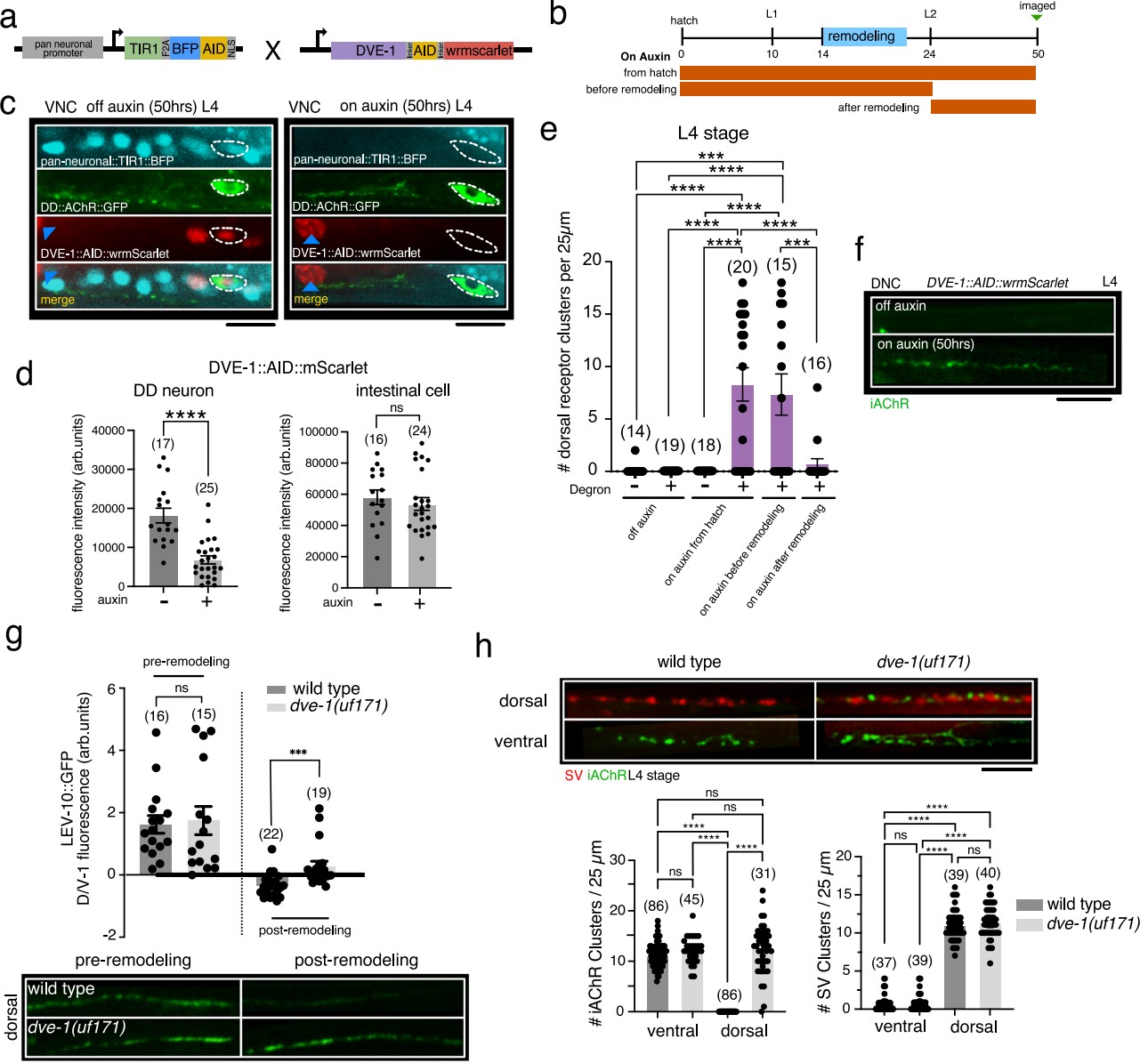

**Fig. 2 | Neuronal DVE-1 is required developmentally for removal of juvenile iAChRs. a** Schematic of *dve-1(syb7515)* [*dve-1::AID::mScarlet*] crossed with reSi7 [*pan-neuronalpr::TIR1::BFP::AID*]. **b** Timelines of auxin treatments for DVE-1 degradation. **c** Ventral nerve cord (VNC) images of L4 animals showing *DVE-1::AID::mScarlet* (red), *flp-13pr::ACR-12::GFP* (green), and *TIR1::BFP::AID* (blue) either under control conditions (left) or with continuous auxin treatment from hatch (right). White dashed circle indicates DD1 cell body. Blue arrowhead, intestinal cell. Scale bar, 5 μm. **d** Quantification of panel (**c**), scatterplot of average *DVE-1::AID::mScarlet* fluorescence in DD1 neurons (left) or intestinal cells (right). Each point represents a single animal. Bars indicate mean ± SEM. ****$p < 0.0001$, ns: not significant, two-tailed students *t*-test. **e** Quantification of DD neuron iAChR clusters in the L4 stage dorsal nerve cord for control and *dve-1::AID::mScarlet* animals under treatment conditions described in panel (**b**). Each point represents a single animal. Bars indicate mean ± SEM., ****$p < 0.0001$, ***$p < 0.001$, two-way ANOVA with Tukey's multiple comparison. **f** Confocal images of ACR-12::GFP clusters in the dorsal nerve cord of L4 stage *dve-1::AID::mScarlet* animals either under control conditions or

with continuous auxin treatment. Scale bar, 5 μm. **g** Top, scatterplot of LEV-10::GFP dorsal/ventral fluorescence intensity ratio measurements per corresponding 25 μm regions of dorsal and ventral nerve cord expressed as dorsal/ventral fluorescence ratio −1. Bars indicate mean ± SEM. ***$p < 0.001$, ns: not significant two-tailed student's *t*-test. Bottom, confocal images of LEV-10::GFP from the dorsal nerve cord before (pre-remodeling) and after (post-remodeling) the L1/L2 transition. Scale bar, 5 μm. NATF DD LEV-10 indicates tissue-specific labeling of endogenous LEV-10 by split-GFP[36]. Each point represents a single animal. **h** Top, confocal images of the dorsal and ventral process from L4 stage wild-type and *dve-1(uf171)* mutants co-expressing the mCherry::RAB-3 synaptic vesicle (SV) and ACR-12::GFP (iAChR) markers in DD neurons. Scale bar, 5 μm. Bottom, quantification of iAChR clusters (left) and SV puncta (right) in dorsal and ventral processes of L4 stage DD neurons for wild-type and *dve-1(uf171)* mutants. Each point represents a single animal. Bars indicate mean ± SEM. ****$p < 0.0001$, ns: not significant, two-way ANOVA with Tukey's multiple comparison test.

the synaptic vesicle marker SNB-1::GFP was increased in dorsal cholinergic axons of L4 stage *dve-1* mutants compared to wild type (Fig. 3k), whereas SNB-1::GFP fluorescence intensity in ventral cholinergic axons of *dve-1* mutants was unchanged (Fig. 3l). We made a similar observation for the mCherry::RAB-3 synaptic vesicle marker

(Fig. S3.1H). Since the DA/B cholinergic neurons form dyadic synapses with both GABA neurons and muscles as postsynaptic targets, we also examined the apposition of cholinergic SVs with AChRs located in postsynaptic muscle cells and found no appreciable difference with wild type (Fig. S3.1I). Notably, the fluorescence intensity of

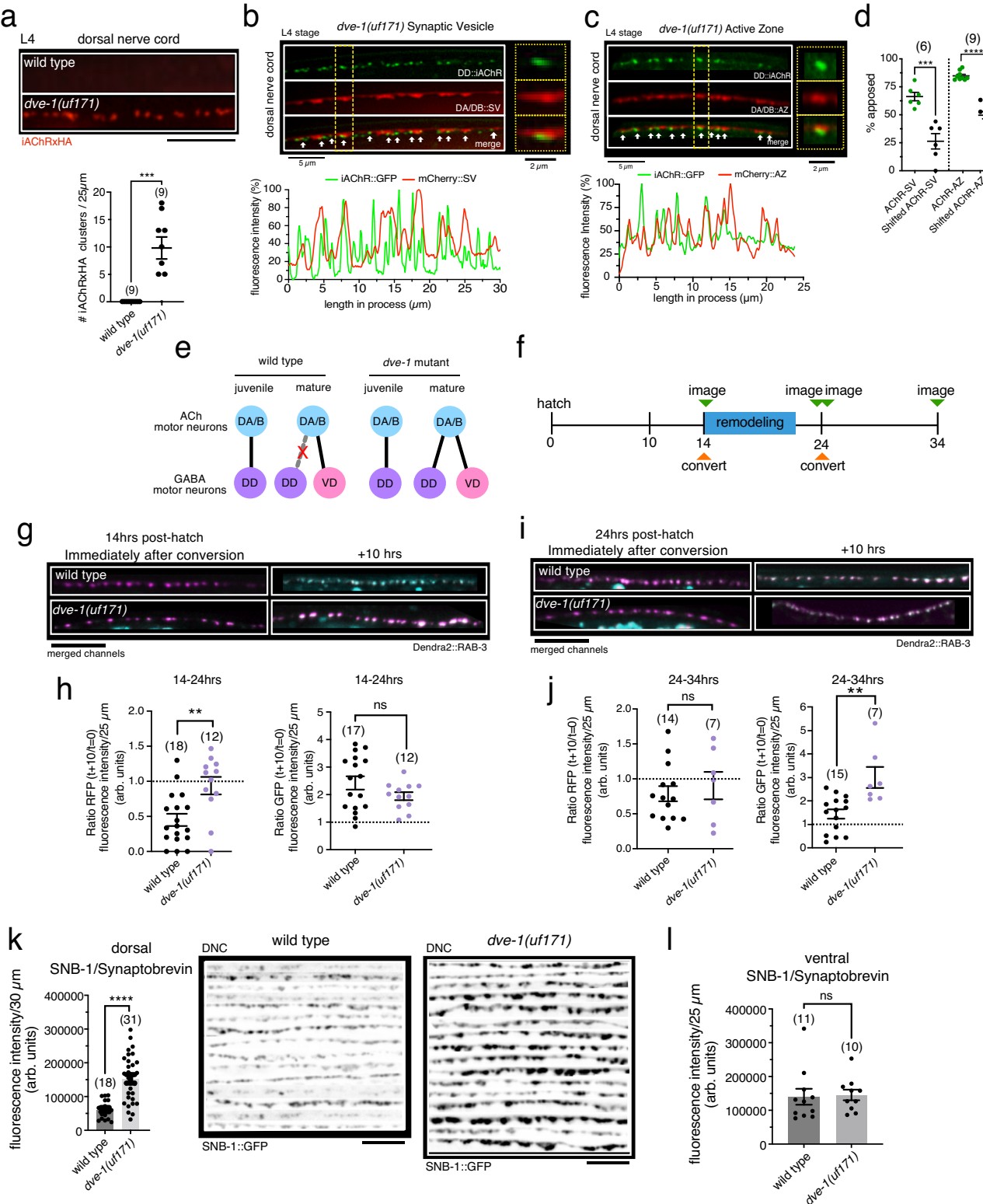

active zone markers UNC-10::GFP and ELKS-1::GFP in dorsal cholinergic axons was also not appreciably altered in L4 stage *dve-1* mutants (Fig. S3.1J, K). Thus, mutation of *dve-1* leads to an increase in the stability or recruitment of synaptic vesicle material at dorsal cholinergic axon terminals but does not appreciably alter the size or density of active zones. Together with our previous findings, these data suggest that DVE-1 promotes destabilization of both vesicle assemblies in presynaptic cholinergic axons and cholinergic postsynaptic sites in GABAergic neurons during wild-type remodeling.

## A failure to eliminate postsynaptic sites leads to enhanced activity and altered motor behavior

We next sought to investigate how a failure of synapse elimination may impact circuit function. We first asked whether the preserved structural connections between dorsal cholinergic axons and GABAergic DD neurons of *dve-1* mutants were functional in adults. To address this question, we used combined cell-specific expression of Chrimson for cholinergic depolarization[40,41] and GCaMP6 for monitoring [$Ca^{2+}$] changes in the postsynaptic GABAergic motor neurons[42] (Fig. S4.1A). We

**Fig. 3 | Mutation of *dve-1* enhances the stability of cholinergic presynaptic sites.**
**a** Top, cell surface iAChR clusters in the L4 stage dorsal nerve cord labeled by anti-HA fluorescence (red). Scale bar, 5 μm. Bottom, scatterplot of dorsal receptor clusters per 25 μm. Each point represents a single animal. Bars indicate mean ± SEM. ***$p < 0.001$, two-tailed students *t*-test. **b** Top, lingering clusters of juvenile postsynaptic iAChRs in DD neurons (green) and apposed cholinergic synaptic vesicles (SV, *unc-129pr*::mCherry::RAB-3, red) from L4 stage *dve-1* mutant dorsal nerve cord. Yellow dash indicates area of inset (right). Bottom, line scan shows relative fluorescence intensity of iAChR (green) and SV (red) for the same 30 μm region. **c** Top, lingering clusters of juvenile postsynaptic iAChRs in DD neurons (green) and presynaptic cholinergic active zone (AZ) marker ELKS-1 (red, *unc-129pr*::ELKS-1::mCherry) from L4 stage *dve-1* mutant dorsal nerve cord. Yellow dash indicates area of inset (right). Bottom, line scan shows relative fluorescence intensity of iAChR (green) and AZ (red) for the same 25 μm region. **d** The percent apposition between postsynaptic iAChR clusters in DD motor neurons and presynaptic cholinergic SVs or AZs (green). As control, each line scan was shifted by 2 μm to assess percent apposed by chance (black). Each point represents a single animal. Bars indicate mean ± SEM. ****$p < 0.0001$, ***$p < 0.001$, two-tailed student's *t*-test.
**e** Synaptic connectivity in the juvenile and mature circuit of wild-type (left) and *dve-1* mutants (right). DA/B, dorsally projecting cholinergic motor neurons. **f** Schematic of photoconversion experiments. **g** Merged red/green image of wild-type (top) and *dve-1(uf171)* mutants (bottom) expressing Dendra2::RAB-3 immediately following photoconversion at 14 h after hatch (left) and 10 h later (right). Red channel, magenta. Green channel, cyan. Red fluorescence indicating juvenile SV clusters

decreases significantly during wild-type remodeling but remains in *dve-1* mutants. Scale bar, 5 μm. **h** Scatterplot showing red (left) and green (right) Dendra2::RAB-3 fluorescence intensity 10 h following photoconversion normalized to the fluorescence intensity immediately after photoconversion at 14 h after hatch (before remodeling) for wild-type (left) and *dve-1(uf171)* mutants (right). Each point indicates a single animal. Bars indicate mean ± SEM. **$p < 0.01$, two-tailed student's *t*-test. **i** Merged red/green image of wild-type (top) and *dve-1(uf171)* mutants (bottom) expressing Dendra2::RAB-3 immediately following photoconversion at 24 h after hatch (left) and 10 h later (right). Red channel, magenta. Green channel, cyan. Scale bar, 5 μm. **j** Scatterplot showing red (left) and green (right) Dendra2::RAB-3 fluorescence intensity 10 h following photoconversion normalized to fluorescence intensity immediately after photoconversion at 24 h after hatch (after remodeling) for wild-type (left) and *dve-1(uf171)* mutants (right). Each point indicates a single animal. Bars indicate mean ± SEM. **$p < 0.01$, ns: not significant, two-tailed student's *t*-test. **k** Left, scatterplot of cholinergic SNB-1::GFP fluorescence intensity in L4 stage dorsal nerve cord (DNC) (*acr-5pr*::SNB-1::GFP) for wild-type and *dve-1* mutants. Each point indicates a single animal. Bars indicate mean ± SEM. ****$p < 0.0001$, two-tailed student's *t*-test. Right, stacked images showing cholinergic SNB-1::GFP clusters in L4 stage dorsal nerve cord of wild-type and *dve-1(uf171)* mutants. Each line is from a different animal. Scale bar, 5 μm. **l** Scatterplot of cholinergic SNB-1::GFP fluorescence intensity in L4 stage ventral nerve cord (*acr-5pr*::SNB-1::GFP) for wild-type and *dve-1* mutants. Each point represents a single animal. Bars indicate mean ± SEM. ns: not significant, two-tailed student's *t*-test.

---

recorded Ca²⁺ transients from young adult GABAergic DD or VD motor neurons in response to presynaptic DA/B cholinergic depolarization. We found that cholinergic photostimulation elicited a modest Ca²⁺ response in roughly 37% of wild-type DD neurons tested, consistent with a low degree of synaptic connectivity between these neurons in adults as predicted by the wiring diagram[15,43]. The percentage of responsive DD neurons (85%) and the average magnitude of stimulus-elicited Ca²⁺ transients were significantly greater in *dve-1* mutants (Fig. 4a), demonstrating enhanced functional connectivity between dorsal cholinergic neurons and GABAergic DD neurons of adult *dve-1* mutants.

We next asked how altered functional connectivity in the motor circuit of *dve-1(uf171)* mutants might affect locomotory behavior. Automated tracking of single worms during exploratory behavior showed *dve-1(uf171)* mutants frequently move in loose, dorsally directed circles, whereas wild-type animals are more likely to adopt straight trajectories (Fig. 4b). During 5 min of continuous tracking, roughly 80% of *dve-1(uf171)* mutants circled or curved, ~60% of these in the dorsal direction, while only 20% of wild-type circled (Fig. 4b, c). The dorsal circling behavior of *dve-1* mutants suggested that altered synaptic output from the motor circuit may produce a turning bias. Mature, wild-type dorsally directed DA/B cholinergic motor neurons form dyadic synapses with dorsal body wall muscles and VD GABAergic dendrites. Based on our mutant analysis, we predict that mature DA/B neurons of *dve-1* mutants preserve additional ectopic dorsal connections with DD GABAergic dendrites, leading to increased cholinergic SV material in the cholinergic DA/B axons. We speculated that the increased abundance of cholinergic synaptic vesicles in dorsal motor axons of *dve-1* mutants may enhance cholinergic activation of dorsal muscles and elicit more robust dorsal turning. In support of this idea, we found that *dve-1* mutants were hypersensitive to the paralyzing effects of the acetylcholinesterase inhibitor aldicarb, an indicator of elevated acetylcholine release[44] (Fig. S4.1B).

To explore this further, we tracked animals during depolarization of dorsal cholinergic neurons by cell-specific photoactivation using Chrimson. Prior to stimulation, control animals moved in predominantly forward trajectories (Fig. 4d). As expected, photostimulation of DA/B motor neurons (625 nm, 14 mW/cm²) enhanced dorsal turning in control animals, often leading to large dorsally oriented circles (Fig. 4d, g). DA/B motor neuron photostimulation elicited heightened turning responses in *dve-1* mutants, increasing dorsal turns by ~2.5 fold compared with photostimulation of controls

and leading to tight dorsally oriented circles (Fig. 4d–h). The enhanced dorsal turning of *dve-1* mutants was associated with an increase in the depth of dorsal bends compared to wild type (Fig. 4g, h, Fig. S4.1C, D) and was not observed in the absence of the chromophore retinal (Fig. S4.1E, F). Chrimson expression was also not appreciably different across *dve-1* mutants and controls (Fig. S4.1G). As DA/B cholinergic motor neurons are presynaptic to both GABAergic neurons and dorsal body wall muscles, we propose that increased acetylcholine release enhances dorsal muscle bending and circling in *dve-1* mutants, perhaps due to an increase in the size of the synaptic vesicle pool in dorsal cholinergic axons. Ectopic activation of dorsally projecting GABAergic DD neurons in *dve-1* mutants might be expected to enhance dorsal inhibition, countering the effects of dorsal excitation. However, the number and strength of synaptic connections from dorsal cholinergic motor neurons to dorsal body wall muscles may overwhelm any increase in dorsal inhibition. Together, our results suggest mutation of *dve-1* impacts functional connectivity both through retention of juvenile connectivity onto DD motor neurons and through an increase in cholinergic transmission onto dorsal muscles. However, we note that alternative models are also possible, such as decreased release from GABAergic DD presynaptic terminals of *dve-1* mutants.

## Synapse elimination occurs through a convergence of DVE-1-regulated destabilization and removal of OIG-1 antagonism

The timing of DD neuron remodeling is, in part, determined through temporally controlled expression of the Ig-domain protein OIG-1. Expression of the transcriptional reporter *oig-1pr*::GFP in L1 stage DD neurons is not appreciably changed in *dve-1* mutants (Fig. 5a), in alignment with prior evidence that other pathways control *oig-1* expression[23,24]. Likewise, *oig-1* deletion does not appreciably alter DVE-1::GFP expression (Fig. 5a). OIG-1 is an Ig domain protein that normally antagonizes synaptic remodeling. In *oig-1* mutants, the remodeling of postsynaptic sites in DD neurons occurs precociously compared with wild type, including both the elimination of dorsal juvenile postsynaptic sites and the formation of new ventral postsynaptic sites[23,24]. While juvenile dorsal postsynaptic sites are removed precociously in *oig-1* single mutants, they are preserved in *oig-1;dve-1* double mutants through late L4 stage, similar to *dve-1* single mutants (Fig. 5b, c, f). DVE-1 is therefore required for synapse elimination in both wild-type and *oig-1* mutants where antagonistic processes promoting synapse stabilization are disrupted. Conversely, new ventral postsynaptic sites are

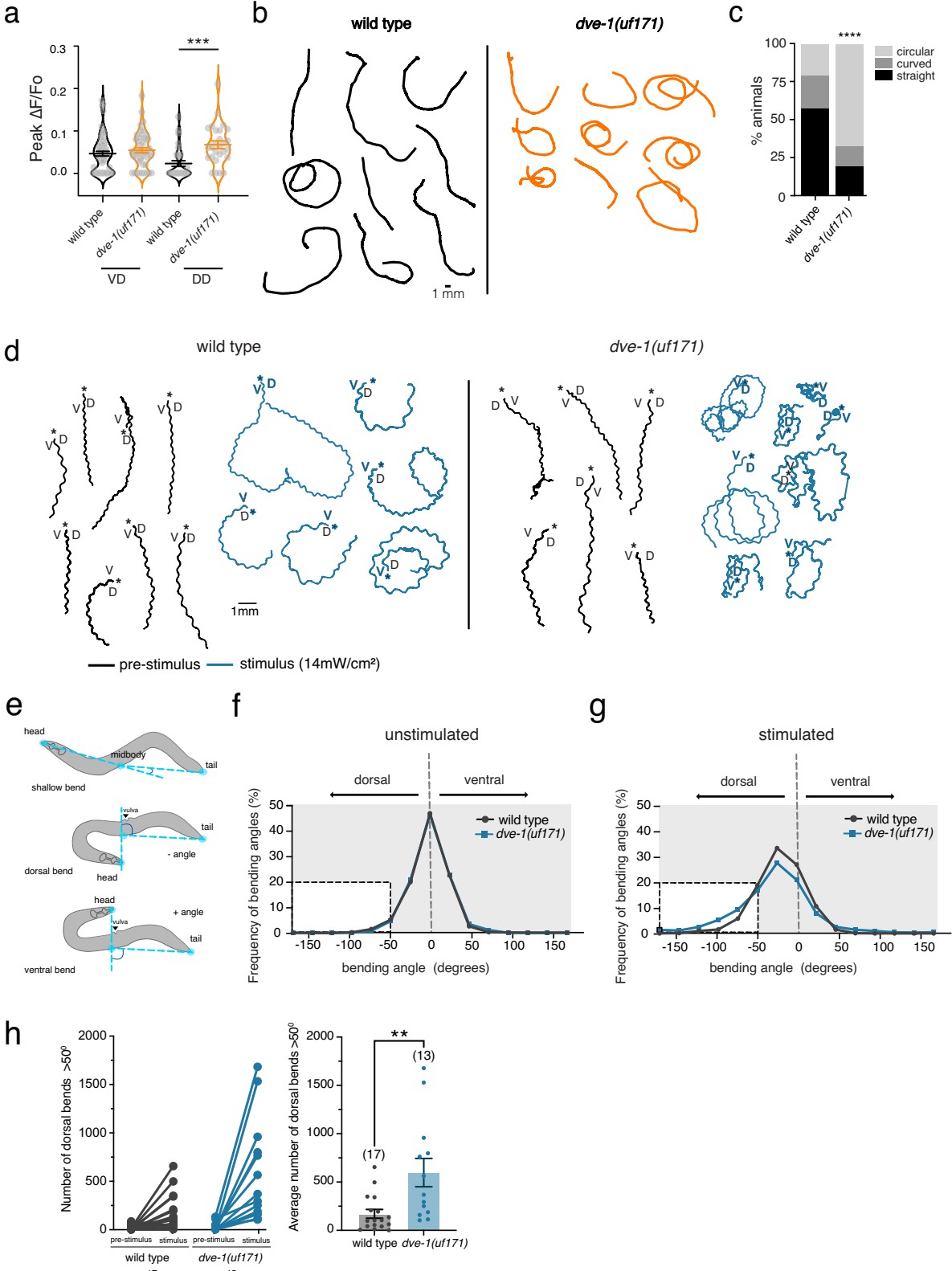

formed precociously in both *oig-1* single mutants and *oig-1;dve-1* double mutants (Fig. 5d–f). Thus, disruption of *dve-1* function reverses precocious synapse elimination in *oig-1* mutants but does not impact the premature assembly of ventral postsynaptic sites, supporting the independence of programs for synapse elimination versus growth, and suggesting independent functions for OIG-1 in each (Fig. 5f). Overall, our findings show that mature connectivity is sculpted through a

convergence of DVE-1 regulated elimination processes and temporally regulated OIG-1 based stabilization mechanisms.

## Nuclear localization of DVE-1 in GABAergic neurons is required for synapse elimination

We next used an engineered *dve-1::gfp* allele[45] to investigate potential mechanisms for DVE-1 spatial regulation in GABAergic neurons. As was

**Fig. 4 | A failure of synapse elimination in *dve-1* mutants produces dorsal turning bias. a** Scatter plot showing peak calcium response ($\Delta F/F_0$) in DD and VD GABAergic neurons to photostimulation of presynaptic DA and DB cholinergic neurons for wild-type and *dve-1(uf171)* mutants. Horizontal bars indicate mean peak $\Delta F/F_0 \pm$ SEM. Non-responders are included as zero values. ***$p < 0.001$, one-way ANOVA with Tukey's multiple comparison. $n = 16$ animals for each condition. Number of cells quantified: wt DD: 30, *dve-1(uf171)* DD: 27, wt VD: 64, *dve-1(uf171)* VD: 47. **b** Representative locomotion tracks for wild-type (black) and *dve-1(uf171)* (orange) animals recorded over 5 min of single worm tracking on NGM OP50 plates. Scale bar, 1 mm. **c** Percentage of straight, curved, or circling tracks for wild-type and *dve-1(uf171)* mutants. ****$p < 0.0001$, two-tailed Chi-square test. wt: $n = 14$, *dve-1(uf171)* $n = 15$. **d** Tracks for wild-type (left) and *dve-1* mutant (right) animals during forward runs (30 s) prior to or during photostimulation. Asterisks indicate the start of the track. D/V indicates dorsal and ventral directions. **e** Schematics of bending angle measurements. Solid orange circles indicate the vertices (head, midbody, and tail) of the body bending angle (blue) measured. **f** Frequency distribution of body bending angles measured prior to photosimulation for wild-type (black) and *dve-1(uf171)* (blue). Negative bending angle values indicate dorsal, while positive bending angle values indicate ventral. Inset highlights bending events >50°. wild-type $n = 17$, *dve-1(uf171)* $n = 13$. **g** Frequency distribution of body bending angles measured during photostimulation for wild-type (black) and *dve-1(uf171)* (blue). Negative bending angle values indicate dorsal, while positive bending angle values indicate ventral. Inset highlights bending events >50°. wild type $n = 17$, *dve-1(uf171)* $n = 13$. **h** Left, scatterplot of total number of dorsal bends >50° before and after photostimulation. Points with connecting lines represent a single animal. Right, average number of dorsal bends >50° during the period of photostimulation for wild-type and *dve-1(uf171)* animals. Bars indicate mean ± SEM. **$p < 0.01$, two-tailed student's *t*-test. wild type $n = 17$, *dve-1(uf171)* $n = 13$.

observed for the *dve-1::AID::wrmScarlet* allele described above, we noted strong DVE-1::GFP expression in intestinal cells, and in roughly 20 neurons at the L1 stage including DD GABAergic neurons (Fig. 6a–c; Fig. S6.1A). Notably, DVE-1::GFP was specifically localized to DD GABAergic nuclei at L1 stage, where it assembled in discrete nuclear foci during the time frame of synaptic remodeling. Similar nuclear DVE-1 clusters were noted previously in intestinal cell nuclei where DVE-1 is thought to regulate gene expression during the mitochondrial unfolded protein response (mtUPR) by associating with loose regions of chromatin and organizing chromatin loops[45]. We found that DVE-1::GFP expression in GABAergic neurons required the Pitx family homeodomain transcription factor UNC-30, the terminal selector of *C. elegans* GABAergic motor neuron identity (Fig. S6.1B, C)[46–48]. Mutation of *unc-30* did not appreciably change DVE-1::GFP fluorescence in intestinal cells (Fig. S6.1D), suggesting cell type-specific mechanisms for *dve-1* expression mediated at least in part through UNC-30 regulation. In support of this idea, mutation of putative UNC-30 binding sites[46–48] identified in the *dve-1* promoter region strikingly reduced *dve-1* expression in GABAergic neurons but had no effect on intestinal *dve-1* expression (Fig. S6.1E).

Prior studies of DVE-1 in intestinal cells showed that deSUMOylation of DVE-1, mediated by the isopeptidase ULP-4, is required for its nuclear localization[49]. We asked if ULP-4 is similarly required for DVE-1 nuclear localization and synapse elimination in DD GABAergic neurons. Mutation of *ulp-4* caused a striking decrease of nuclear DVE-1::GFP fluorescence in GABAergic neurons and severely diminished DVE-1::GFP nuclear foci (Fig. 6b, c). A mutated form of DVE-1::GFP, DVE-1K327R, where a key lysine residue required for SUMOylation is mutated to arginine[49], localizes to GABAergic nuclei in the absence of *ulp-4* (Fig. 6b, c). ULP-4 was also required for the elimination of dorsal iAChR clusters during remodeling, such that dorsal iAChR clusters remained present in roughly 50% of L4 stage *ulp-4* mutants (Fig. 6d–f). Either pan-neuronal or GABA neuron-specific expression of the wild-type *ulp-4* gene in *ulp-4* mutants was sufficient to restore the elimination of dorsal iAChRs, while intestinal *ulp-4* expression was not (Fig. 6d–f). Further, mutation of the DVE-1 SUMOylation site (K327R) by itself did not impair synapse elimination but restored proper removal of iAChRs in *ulp-4* mutants (Fig. 6d–f). We conclude that the localization of DVE-1 to GABAergic nuclei is essential for synapse elimination during remodeling, and this localization is regulated at least in part by ULP-4 and SUMOylation. Notably, the nuclear localization of mammalian SATB family members is also dependent on SUMOylation, suggesting conserved regulatory mechanisms[50,51].

### Analysis of potential DVE-1 transcriptional targets reveals several pathways with relevance for synapse elimination

Recent work revealed that homeodomain transcription factors are broadly utilized in the specification of *C. elegans* neuronal identity[34]. Given this finding and DVE-1 homology with mammalian SATB family transcription factors, we asked whether DVE-1 transcriptional regulation may be important for GABAergic neuronal identity. We found that the numbers of DD neurons and commissures were unchanged in *dve-1* mutants compared to wild type (Fig. S6.2A). In addition, we found that the expression levels for *oig-1* (Fig. 5a) and three additional GABAergic markers in DD neurons (*unc-47*/GABA vesicular transporter, *unc-25*/glutamic acid decarboxylase, and *flp-13*/FMRFamide neuropeptide) were not appreciably altered by *dve-1* mutation (Fig. S6.2B). These results support that DVE-1 is not critical for GABAergic identity of the DD neurons but instead regulates other aspects of GABAergic neuron development.

To reveal potential direct targets of DVE-1 in GABAergic neurons, we analyzed chromatin immunoprecipitation followed by sequencing (ChIP-Seq) data available from the modENCODE consortium[52]. We found 1044 genes with strong DVE-1 binding signal in their promoter regions, implicating these genes as potential direct targets for DVE-1 transcriptional regulation (Supplementary Data 5), though we note that overexpression of DVE-1::GFP in the strain used for these experiments could potentially impact this analysis. Through de novo motif discovery analysis, we identified four sequences overrepresented in the DVE-1 binding peaks, two of which were identified previously (Fig. S7.1A)[53]. 627 of the identified potential DVE-1 targets are significantly expressed in GABAergic neurons based on available single-cell RNA-seq data (Supplementary Data 5)[54,55]. Pathway analysis of the GABAergic neuron-enriched targets using WormCat[56] and WormenrichR[57] revealed a significant enrichment of genes involved in the mitochondrial unfolded protein response (mtUPR) stress pathway (Fig. 7a, Fig. S7.1B, Supplementary Data 5), as expected from prior studies[31,49,58]. Notably, our analysis also revealed enrichment of genes involved in the ubiquitin-proteasome system (UPS), as well as various other processes including ribosomal composition, and endocytic and phagocytotic function (Fig. 7a, Fig. S7.1B, Supplementary Table 1, Supplementary Data 5) that represent intriguing potential targets for DVE-1 regulation of synapse elimination.

### Inhibition of the ubiquitin-proteasome system, but not activation or inhibition of the mtUPR, delays DVE-1-dependent synapse elimination

To assess which of the pathways identified from our analysis may be most critical, we next asked whether DVE-1 regulates synapse elimination by activating or inhibiting the mtUPR. We first quantified the length and density of mitochondria in DD neurons and found no differences between wild-type and *dve-1* mutants (Fig. S7.2A). We next measured mtUPR activation in *dve-1* mutants by quantifying the fluorescence of *hsp-6pr*::GFP, a commonly used mtUPR reporter[31]. Surprisingly, we noted increased levels of intestinal *hsp-6pr*::GFP expression in *dve-1* mutants compared with control, suggesting elevated mtUPR activity (Fig. S7.2B, C). The transcription factor ATFS-1 is required for the *hsp-6pr*::GFP transcriptional response and activation

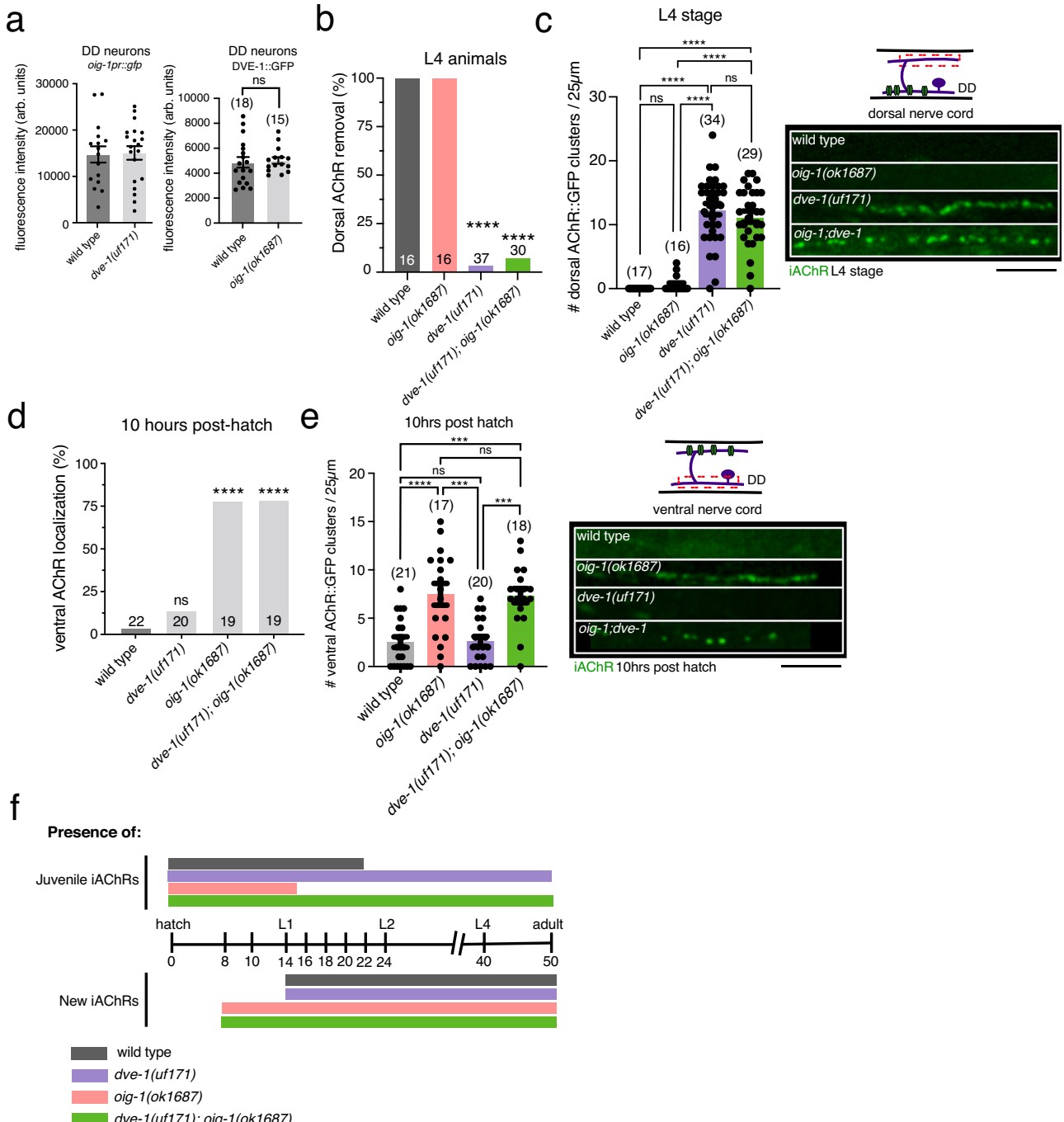

**Fig. 5 | DVE-1-regulated pathways for synapse destabilization act in parallel to _oig-1_ antagonism of remodeling. a** Left, average _oig-1pr::gfp_ fluorescence intensity in DD soma of L1 stage wild-type and _dve-1_ mutants. Each point represents a single DD cell body. Imaged 2 DD neurons/animal. Wild type _n_ = 8, _dve-1_ mutants _n_ = 10. Right, average nuclear DVE-1::GFP fluorescence intensity in DD neurons of L1 stage wild-type and _oig-1(ok1687)_ mutants. Each point represents a single animal. Bars indicate mean ± SEM. ns: not significant, two-tailed student's _t_-test. **b** The percentage of animals where dorsal iAChRs are eliminated for L4 stage wild-type, _oig-1(ok1687)_, _dve-1(uf171)_, and _dve-1(uf171);oig-1(ok1687)_ double mutants. ****_p_ < 0.0001, two-tailed Fischer's exact test with Bonferroni Correction. **c** Left, quantification of average number of iAChR clusters in L4 stage DD neurons per 25 μm of the dorsal nerve cord for the genotypes indicated. Each dot represents a single animal. Bars indicate mean ± SEM. ****_p_ < 0.0001, ns: not significant, one-way ANOVA with Tukey's multiple comparisons test. Right, fluorescent confocal images of iAChR clusters in L4 stage DD neurons of the dorsal nerve cord for the genotypes indicated. Scale bar, 5 μm. **d** The percentage of animals where iAChRs are localized to the ventral side at 10 h post hatch for wild-type, _dve-1(uf171)_, _oig-1(ok1687)_, and _dve-1(uf171);oig-1(ok1687)_ double mutants. ****_p_ < 0.0001, two-tailed Fischer's exact test with Bonferroni Correction. **e** Left, quantification of average number of iAChR clusters in DD neurons per 25 μm of the ventral nerve cord at L1 stage for the genotypes indicated. Each dot represents a single animal. Bars indicate mean ± SEM. ****_p_ < 0.0001, ***_p_ < 0.001, ns: not significant, one-way ANOVA with Tukey's multiple comparisons test. Right, fluorescent confocal images of iAChR clusters in the ventral processes of L1 stage DD neurons. Scale bar, 5 μm. **f** Timeline of development, approximate timing of transitions between larval stages and to adulthood are indicated. Bars indicate the presence of juvenile dorsal iAChRs (top) or ventral iAChRs formed during remodeling (bottom). wild-type (gray), _dve-1_ mutants (purple), _oig-1_ mutants (pink), _oig-1;dve-1_ double mutants (green).

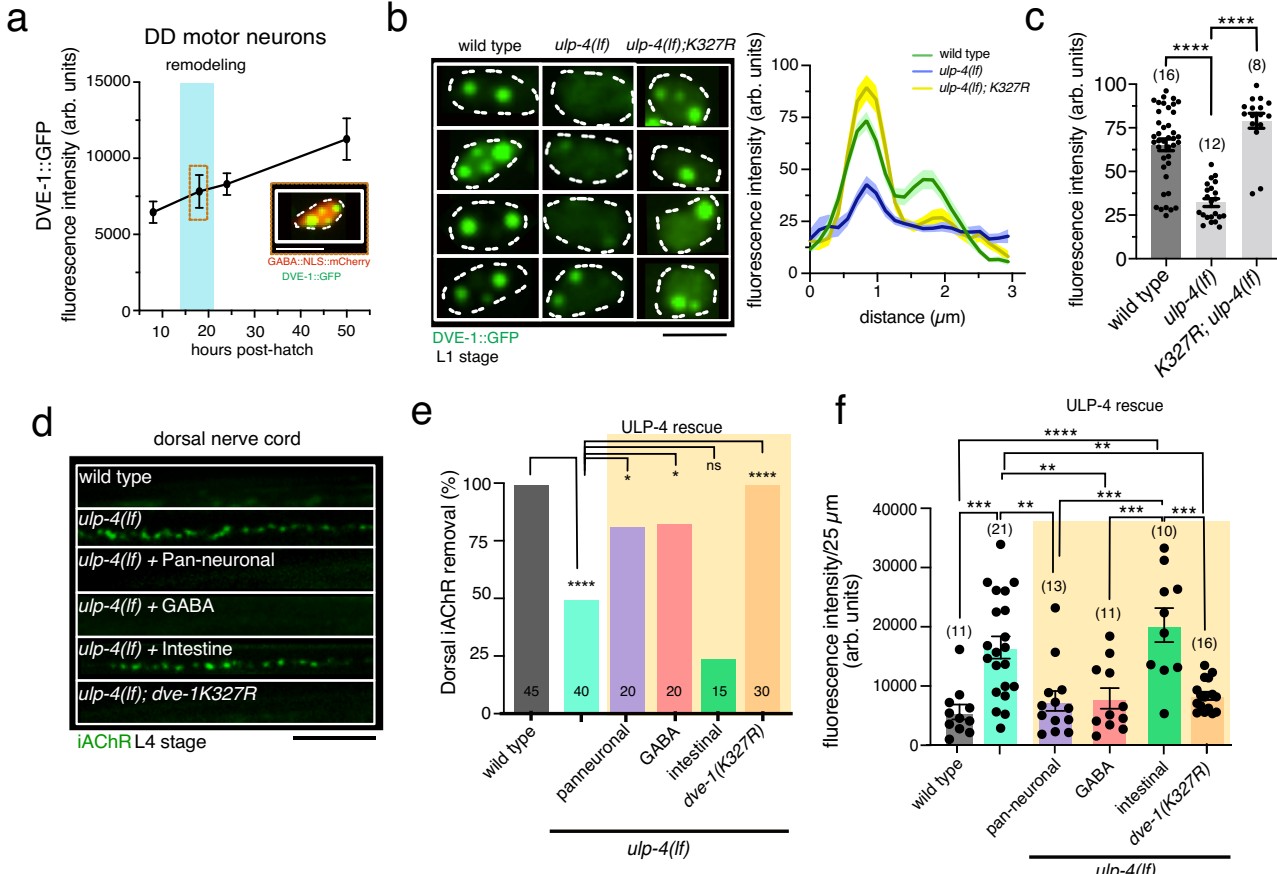

**Fig. 6 | The deSUMOylating peptidase ULP-4 regulates DVE-1 nuclear localization. a** Fluorescence intensity of nuclear DVE-1::GFP in DD motor neurons at 10, 18, 22 and 50 h after hatch. Nuclear DVE-1::GFP is organized in discrete foci and increases through development. 10 h $n = 10$, 18 h $n = 7$, 22 h $n = 8$, 50 h $n = 9$. Each time point indicates mean ± SEM from at least three independent experiments. Inset, representative image of nuclear DVE-1::GFP in DD motor neuron 18 h after hatch (nucleus labeled by NLS::mCherry). White dashed line outlines nucleus. Scale bar, 3 µm. **b** Confocal fluorescence images (left) of DVE-1::GFP in DD GABAergic motor neurons of L1 stage wild-type, *ulp-4(lf)* mutant, and *K327R;ulp-4(lf)* double mutants. *ulp-4(lf)* disrupts DVE-1::GFP nuclear localization. White dashed line outlines nucleus. Scale bar, 3 µm. Right, line scan of nuclear DVE-1::GFP fluorescence intensity in DD motor neurons of wild-type (green) $n = 16$, *ulp-4(lf)* (blue) $n = 11$ and DVE-1(K327R);*ulp-4(lf)* double mutants (yellow) $n = 8$. Solid line represents mean, shading represents standard deviation of fluorescence. **c** Scatterplot (left) of the peak nuclear DVE-1::GFP fluorescence intensity in DD motor neurons. Each point

represents a single DD nucleus. Imaged at least 2 DD cells per animal at L1 stage. Wild type: $n = 16$, *ulp-4(lf)*: $n = 11$, *K327R;ulp-4(lf)*: $n = 8$. Bars indicate mean ± SEM. ****$p < 0.0001$, one-way ANOVA with Tukeys multiple comparisons test. **d** Fluorescent confocal images of iAChR clusters in the dorsal processes of L4 stage DD neurons for genotypes indicated. Scale bar, 5 µm. **e** Quantification of iAChR clustering at L4 stage for the genotypes indicated. Bars indicate the percentage of L4 stage animals where dorsal iAChRs have been completely removed. ****$p < 0.0001$, *$p < 0.05$, ns: not significant, two-tailed Fischer's exact test with Bonferroni correction. 1/2 pan-neuronal lines, 2/2 GABA lines, and 0/2 intestinal rescue lines restored proper removal of dorsal iAChRs by L4. **f** Scatterplot of average iAChR fluorescence intensity per 25 µm in the dorsal nerve cord at L4 stage for the genotypes indicated. Each point represents a single animal. Bars indicate mean ± SEM. ****$p < 0.0001$, ***$p < 0.001$, **$p < 0.01$, one-way ANOVA with Tukey's multiple comparisons test.

of the mtUPR[59]. RNAi knockdown of *atfs-1* decreased *hsp-6pr*::GFP expression in *dve-1* mutants (Fig. S7.2C), indicating that down-regulation of *atfs-1* reduced mtUPR activation. However, inhibition of the mtUPR by *atfs-1* knockdown failed to restore normal removal of dorsal iAChR clusters in *dve-1* mutants (Fig. S7.2D). Likewise, a null mutation in *atfs-1* did not alter synapse elimination in otherwise wild-type animals and failed to restore synapse elimination when combined with mutation of *dve-1* in *atfs-1;dve-1* double mutants (Fig. 7b). These results show that increased activation of the mtUPR in *dve-1* mutants is not sufficient to account for a failure in synapse removal. Consistent with this interpretation, we note that mutation of *ulp-4* suppresses mtUPR activation but disrupts synapse elimination (Fig. 6d–f; Fig. S7.2B). Constitutive mtUPR activation by mutation of the mitochondrial complex III subunit gene *isp-1* also did not alter synapse elimination (Fig. 7b; Fig. S7.2C)[60]. In addition, mutation of *ubl-5*, a cofactor with DVE-1 in the initiation of the intestinal mtUPR[61], did not affect synapse elimination (Fig. 7b). Our findings demonstrate that

mtUPR activation or inhibition do not alter synapse removal. We conclude that DVE-1 coordinates synapse elimination through transcriptional regulation of alternate pathways, perhaps those identified from our enrichment analysis.

Given recent evidence for the regulation of synapse structure through ubiquitin-dependent degradation processes and links with neurological disease[62], we next asked whether DVE-1 control of synapse elimination may occur through transcriptional regulation of the ubiquitin-proteasome system. Using quantitative RT-PCR, we first investigated the expression of selected putative DVE-1 targets involved in UPS function. The expression of three of the four genes we tested (*cul-5*, *eel-1*, *spat-3*) was significantly altered in *dve-1* mutants (Fig. S7.2E), suggesting DVE-1 regulation of their expression. We next investigated involvement of the UPS during synapse elimination using bortezomib, a small molecule inhibitor of the 26S proteasome. Treatment with high (≥10 µM) concentrations of bortezomib produced larval arrest. In contrast, treatment with 5 µM bortezomib disrupted

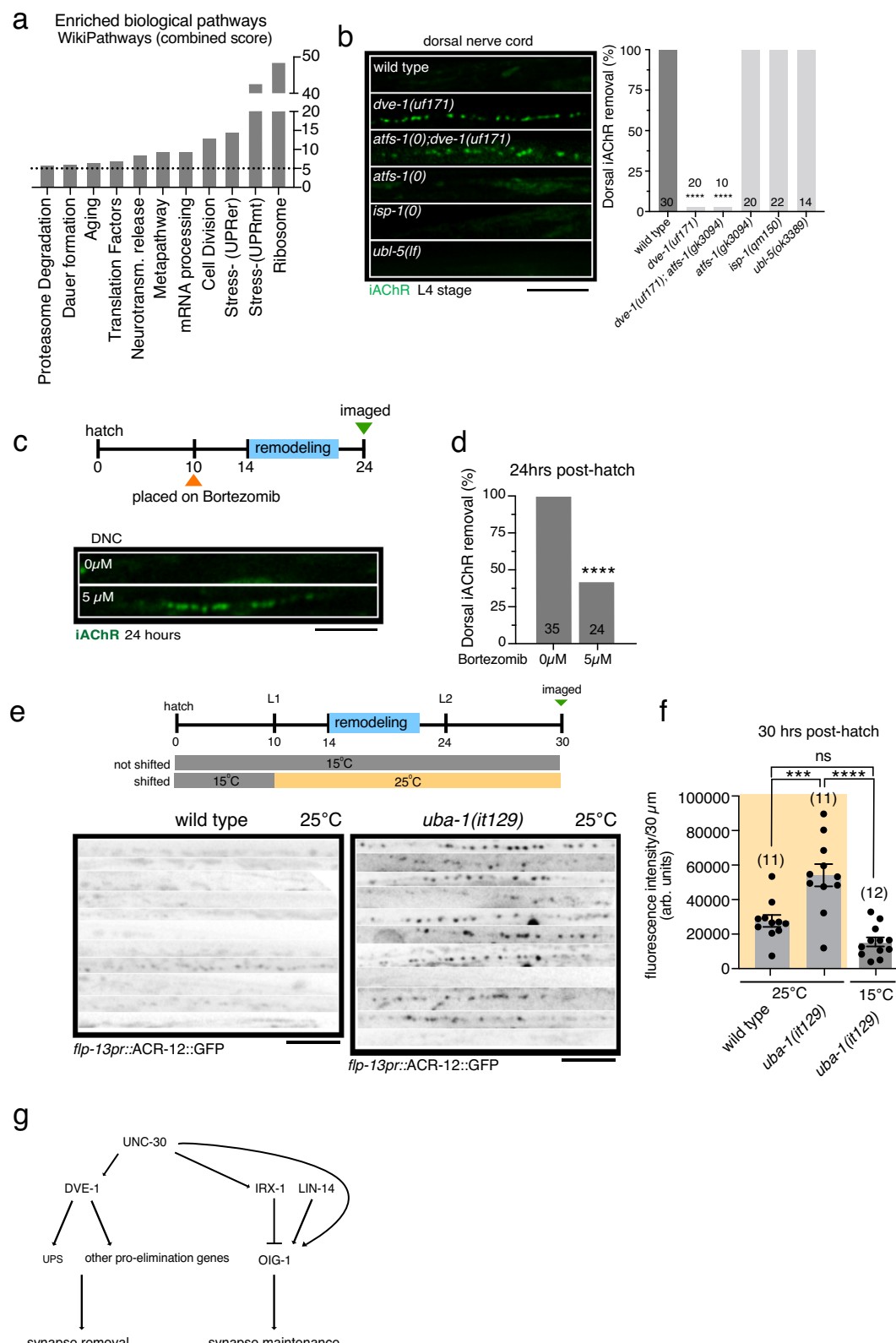

UPS function, as assessed by induction of the *skn-1* dependent proteasome reporter *rpt-3pr*::GFP[63] (Fig. S7.2F), but did not cause developmental arrest or appreciably reduce dendritic spines in mature DD neurons (Fig. S7.2G). Notably, treatment with 5 μM bortezomib significantly delayed synapse elimination during remodeling. More than 50% of animals treated with bortezomib failed to remove juvenile dorsal postsynaptic sites in DD neurons by 24 h after hatch (Fig. 7c, d),

suggesting a requirement for UPS function in synapse elimination. Given the potential for redundant functions amongst the putative UPS DVE-1 targets we identified, we asked if synapse elimination was affected in animals carrying a temperature-sensitive allele of the sole *C. elegans* E1 ubiquitin ligase, *uba-1*[64]. We found that synapse elimination was significantly delayed in *uba-1* mutants shifted to the restrictive temperature at 10 h after hatch (Fig. 7e, f), but was unaffected in

**Fig. 7 | Disruption of the ubiquitin-proteasome system delays synapse elimination. a** WormEnrichR pathway analysis using WikiPathway. Bars represent enriched pathways with a combined score >5 (dashed line) and p-adj < 0.05. **b** Activation or inhibition of mtUPR has no effect on synapse elimination. Left, iAChR clusters in the dorsal processes of DD neurons at L4 stage. Scale bar, 5 μm. Right, bars indicate the percentage of L4 stage animals with dorsal iAChRs eliminated. ****$p$ < 0.0001, two-tailed Fischer's exact test, Bonferroni correction. **c** Top, schematic of Bortezomib inhibitor experimental design. Animals were treated with Bortezomib at 10 h after hatch until imaging at 24 h after hatch. Bottom, iAChR clusters in the dorsal DD neuron processes at 24 h after hatch in control or following bortezomib treatment. Scale bar, 5 μm. **d** The percentage of animals where dorsal iAChRs are removed by 24 h after hatch in control or following bortezomib treatment. ****$p$ < 0.0001, two-tailed Fischer's exact test, Bonferroni correction. **e** Top, schematic of the experimental design for temperature shift experiments. Wild type and *uba-1(it129)* mutants were either maintained at the permissive temperature (15 °C) or moved to the restrictive temperature (25 °C) at 10 hrs after hatch before imaging at 30 h after hatch. Bottom, iAChR clusters in the DD dorsal process of wild-type (right) and *uba-1(it129)* (left) mutant animals shifted to restrictive temperature and imaged at 30 h after hatch. Each line represents the dorsal process of a different animal. Scale bar, 5 μm. **f** The average fluorescence intensity per 25 μm of wild-type or *uba-1(it129)* mutant dorsal nerve cord at the temperature indicated. Bars indicate mean ± SEM. ****$p$ < 0.0001, ***$p$ < 0.001, ns: not significant, one-way ANOVA with Tukey's multiple comparisons test. Each point represents a single animal. **g** Schematic for control of remodeling. The transcription factor UNC-30/Pitx regulates the expression of *dve-1* and *oig-1*[23,24], this paper. *oig-1* encodes an Ig-domain protein that stabilizes juvenile synapses prior to remodeling. Temporal control of *oig-1* expression occurs through LIN-14- and IRX-1-mediated transcriptional regulation[23,24]. DVE-1 promotes synapse removal/destabilization, perhaps through UPS transcriptional regulation. Mutation of *dve-1* impairs synapse removal even when OIG-1-mediated stabilization is disrupted.

animals raised continuously at the permissive temperature. While additional DVE-1 regulated pathways likely contribute, our identification and analysis of UPS pathway genes as potential targets for direct transcriptional regulation by DVE-1 lead us to propose that cell-autonomous DVE-1 transcriptional regulation of the ubiquitin proteasome system may be an important step for synapse elimination during remodeling (Fig. 7g).

## Discussion

Developmental remodeling of synaptic connectivity occurs throughout phylogeny, refining and reorganizing neuronal connections toward the establishment of the mature nervous system. While neuron-extrinsic events that shape remodeling, for example, microglial phagocytosis of synaptic material[5–8], have gained a lot of recent attention, neuron-intrinsic processes governing remodeling have remained less well-described. Likewise, the relationship between degenerative and growth processes during remodeling has not been clearly elucidated. The developmental remodeling of *C. elegans* GABAergic DD neurons presents a uniquely accessible system for addressing important questions about evolutionarily conserved neuron-intrinsic mechanisms of remodeling because the reorganization of their connectivity occurs without gross morphological changes or a requirement for synaptic removal by other cell types.

Here, we show that the homeodomain transcription factor DVE-1 is specifically required for the elimination of juvenile synaptic inputs to DD neurons during remodeling. In *dve-1* mutants, juvenile postsynaptic sites and apposing cholinergic presynaptic terminals are preserved into adulthood. The failure to eliminate these sites results in elevated activity at these synapses and impaired motor function. Interestingly, *dve-1* is not required for the growth of new DD neuron synaptic inputs that are characteristic of the mature motor circuit, indicating that the formation of new connections is not dependent upon the elimination of pre-existing juvenile synapses. Likewise, mutation of *dve-1* does not alter the developmental reorganization of synaptic outputs from DD neurons onto muscles. In *dve-1* mutants, newly relocated GABAergic synaptic terminals occupy similar territories in DD neurons as lingering juvenile synaptic inputs. Thus, the formation of new GABAergic presynaptic terminals during the maturation of the circuit is not contingent on the elimination of nearby juvenile postsynaptic sites in DD neurons.

Our findings lead us to propose that cell-autonomous transcriptional regulation of GABAergic neurons by DVE-1 promotes the elimination of their juvenile synaptic inputs. We found that DVE-1 is expressed in a limited number of neurons, including GABAergic motor neurons, and DVE-1 localization to GABAergic nuclei is required for synapse elimination to proceed normally. DVE-1 regulation of synapse elimination shares interesting parallels with a previously described pathway for the elimination of postsynaptic structures at mouse glutamatergic synapses through transcriptional regulation by Myocyte Enhancer Factor 2 (Mef2)[65,66]. However, in contrast to MEF2-regulated synapse elimination, we found that DVE-1-dependent elimination is not strongly activity-dependent. Also, we did not observe strong temporal regulation of DVE-1 expression in GABAergic neurons prior to or during remodeling, raising important mechanistic questions about the timing of synapse elimination. One possible route for temporal regulation might be through control of DVE-1 nuclear localization. We show that DVE-1 localization to GABAergic nuclei can be regulated through SUMOylation. However, we observed nuclear localization of DVE-1 in GABAergic neurons prior to the onset of remodeling, suggesting the presence of additional mechanisms for temporal control. This is consistent with prior work indicating that temporally controlled expression of OIG-1 regulates the timing of remodeling[23,24]. We found that precocious synapse elimination in *oig-1* mutants is reversed when *dve-1* function is also disrupted, further indicating that DVE-1 transcriptional regulation is required for synapse elimination to occur. In contrast, precocious growth of new synapses in *oig-1* mutants was not altered by mutation of *dve-1*, suggesting that DVE-1-regulated degenerative mechanisms act in parallel with growth processes that are regulated independently.

Our experiments show that mutation of *dve-1* affects the stability of both postsynaptic sites in DD GABAergic neurons and presynaptic terminals in cholinergic neurons. We show that the juvenile synaptic vesicle assemblies in axons of presynaptic cholinergic neurons are almost completely exchanged during the 10-h period of wild-type remodeling. This turnover of synaptic vesicles during wild-type remodeling contrasts with their relative stability over the 10 h immediately following remodeling. Notably, the turnover of juvenile cholinergic synaptic vesicle assemblies during remodeling is strikingly reduced in *dve-1* mutants, indicating that disruption of DVE-1 transcriptional activity is sufficient to stabilize presynaptic vesicle pools in cholinergic neurons. Since DVE-1 expression is limited to postsynaptic GABAergic neurons, our results suggest that DVE-1-regulated postsynaptic pathways promote the exchange or elimination of juvenile presynaptic elements through destabilization of the postsynaptic apparatus and associated signaling components. Our photoconversion experiments also show that the recruitment of new synaptic vesicles in cholinergic axons of *dve-1* mutants is not halted by the stabilization of juvenile synaptic vesicle assemblies. We noted an overall increase in dorsal synaptic vesicle material in *dve-1* mutants compared with either wild-type or ventral synapses of *dve-1* mutants. We speculate that the retention of juvenile synaptic vesicle clusters in *dve-1* mutants occurs in parallel with the formation of new synaptic connections between cholinergic DA/B motor neurons and post-embryonic born VD GABAergic neurons. The increased synaptic vesicle material in dorsal cholinergic axons of *dve-1* mutants may therefore arise from the additive effects of these two processes. Our Ca$^{2+}$ imaging and

behavioral experiments provide evidence that the increase in cholinergic synaptic vesicles of dorsally projecting motor neurons alters circuit function such that cholinergic activation of dorsal musculature is enhanced in *dve-1* mutants, resulting in deeper dorsal bends and a dorsal turning bias during movement.

Our pathway analysis of DVE-1 ChIP-seq data showed enrichment of genes involved in the mtUPR. In the mtUPR, DVE-1/SATB is thought to organize loose chromatin to induce the expression of chaperones and proteases[31,58,61]. However, manipulations that either activated or inhibited the mtUPR did not affect remodeling, providing support for a model where DVE-1 regulation of remodeling occurs independently of the mtUPR. Our analysis of potential DVE-1 targets revealed the enrichment of genes in other pathways that may be important for the removal of synapses, in particular UPS pathway genes. Notably, pharmacological inhibition of proteasome function or genetic disruption of *uba-1* caused a striking delay in synapse elimination, supporting the involvement of this pathway and suggesting that DVE-1 transcriptional regulation of the proteasome may be important to promote synapse elimination. We note that potential synaptic targets for direct DVE-1 regulation were also present in the ChIP-seq dataset (Supplementary Data 5), perhaps suggesting several modes of regulation. For example, UNC-40 has been shown to organize synapses through the MADD-4 ligand[67,68] and is important for sexually dimorphic synapse pruning[69].

The closest homolog of DVE-1 is the *Drosophila* homeodomain transcription factor *defective proventriculus* (*Dve*). Interestingly, transcriptional profiling of *Drosophila* mushroom body gamma neurons during their remodeling showed Dve expression peaks at the onset of remodeling (https://www.weizmann.ac.il/mcb/Schuldiner/resources)[70]. DVE-1 also shares homology with the mammalian SATB proteins 1 and 2. SATB transcription factors have roles in many areas of mammalian brain development, such as the activation of immediate early genes important for maintaining dendritic spines in GABAergic interneurons[71] and cortex development and maturation[72], but roles in synapse elimination had been previously uncharacterized. Our findings offer a striking example of DVE-1/SATB transcriptional activation of pro-degenerative pathways acting in concert with temporally controlled expression of a maintenance factor to control a developmentally defined period of synapse elimination. Given the sequence similarities between DVE-1 and mammalian SATB proteins, our analysis may point toward new mechanisms by which SATB family transcription factors control brain development. Importantly, dysfunction of these transcription factors in humans, as in SATB2-associated syndrome, is characterized by significant neurodevelopmental delays, limitations in speech, and severe intellectual disability[25,26]. More broadly, our findings highlight a cellular strategy for temporal control of circuit development through convergent regulation of antagonistic cellular processes. Interestingly, spatiotemporal regulation through competing parallel transcriptional programs is utilized in other developmental contexts across different species[73–75], suggesting this represents a broadly utilized mechanism for temporal control of key developmental events.

## Methods
### Strains and genetics
All strains are derivatives of the N2 Bristol strain (wild type) and maintained under standard conditions at 20–25 °C on nematode growth media plates (NGM) seeded with *E. coli* strain OP50. Some strains were provided by the *Caenorhabditis* Genetics Center (CGC), which is funded by NIH Office of Research Infrastructure Programs (P40 OD010440), and by the National BioResource Project which is funded by the Japanese government. Transgenic strains were generated by microinjection of plasmids or PCR products into the gonad of young hermaphrodites. Integrated lines were produced by X-ray irradiation or UV-integration and outcrossed to wild type/N2 Bristol. A complete list of all strains used in this work is included in Supplementary Data 1.

### Generation of endogenously tagged strains
Strain PHX7515 *dve-1(syb7515*) [*dve-1*::AID::mScarlet] was generated in N2 animals by SunyBiotech. Linker, AID, and mScarlet sequences were inserted after exon 9 at the 3' end of ZK1193.5a.1/*dve-1*. Strain IZ4473 *dve-1(uf206)* was generated in *syb1984* (DVE-1 CRISPR-Cas9-mediated GFP knockin, gifted by the Tian lab) animals. A lysine to arginine (K-R) mutation was created by changing AAA to CGT in exon 6, 4783 bp downstream of start. The IDT CRISPR HDR design tool (https://www.idtdna.com/pages/tools/alt-r-crispr-hdr-design-tool) was used to generate repair templates and guide sequences. Animals were injected with CRISPR/Cas9 mix [crRNA (oligo 2 nmol, IDT), donor (oligo 4 nmol, IDT), purified Alt-R S.p. Cas9 nuclease V3 100 μg (IDT CAT 1081058), Alt-R CRISPR/Cas9 tracrRNA (5 nmol, IDT CAT 1072532), and pRF-4 (*rol-6* plasmid)]. Rolling worms were singled and validated by PCR sequencing. CRISPR/Cas9 design is provided in Supplementary Data 4.

### Molecular biology
Plasmids were constructed using two-slot Gateway Cloning system (Invitrogen) and confirmed by restriction digest and/or sequencing as appropriate. All plasmids and primers used in this work are described in Supplementary Data 2 and 3 respectively.

For *dve-1* genomic rescue, the *dve-1* promoter (5 kb upstream from translational start site), genomic fragment (from translational start to stop, 6107 bp), and *dve-1* 3' UTR (626 bp downstream from translational stop) were amplified from genomic N2 DNA. For *dve-1* GABA-specific rescue, DVE-1 long (ZK1193.a) and short (ZK1193.c) isoform cDNA was amplified from RNA and ligated into SacII digested pCL86 and XbaI digested pCL101, respectfully to generate pDest-178 (*dve-1* cDNAa) and pDest-252 (*dve-1* cDNAc). pDest-178 and pDest-252 were recombined with pENTR-*unc-47* to generate pKA17 (*unc-47pr*::*dve-1*a) and pJR21(*unc-47pr*:: *dve-1*c). pKA17 and pJR21 were injected simultaneously (30 ng/μl).

pKA110 (*unc-129pr*::Dendra2::RAB-3) was created by ligating a gBlock (IDT) containing the Dendra2 coding sequence into NheI-HF/PstI-HF digested pDest-114 to generate pDest-339 (Dendra2::RAB-3). pDest-339 was then recombined with pENTR-*unc-129* to generate pKA110 and injected at 50 ng/μl. To generate pKA35 (*unc-129pr*::Chrimson::SL2::BFP), Chrimson was amplified from pDest-104 and ligated into NheI-HF/BstBI digested pDest-239 to generate pDest-240 (Chrimson::SL2::BFP). pDest-240 was then recombined with pENTR-*unc-129* to generate pKA35 and injected at 50 ng/μl.

To generate *ulp-4* rescue constructs, *ulp-4* cDNA was amplified from RNA and ligated into NheI-HF/KpnI-HF digested pDest-139 to generate pDest-291. pDest-291 was recombined with pENTR-*unc-47* to generate pKA76 (*unc-47pr*::*ulp-4* cDNA), pENTR-F25B3.3 to generate pKA78 (*F25B3.3pr*:: *ulp-4* cDNA), and pENTR-*gly-19* to generate pKA80 (*gly-19pr*:: *ulp-4* cDNA). All *ulp-4* rescue constructs were injected at 30 ng/μl.

pKA74 (*unc-47pr*::NLS::mCherry) was created by amplifying an artificial intron and NLS from plasmid #68120 (Addgene) and was ligated into AgeI-HF/XbaI digested pDest-145 to generate pDest-205 (NLS::mCherry). pDest-205 was recombined with pENTR-*unc-47* to generate pKA74 and injected at 50 ng/μl.

pJR18 (Δ*dve-1pr*::DVE-1::GFP) was generated by ligating a gBlock (IDT) containing the *dve-1* promoter missing six putative UNC-30 binding sites (1214–1224, 1407–1417, 1456–1466, 1597–1607, 1634–1644, and 1782–1792 bp upstream of the *dve-1* start and ligated into PsiI/SacI cut *dve-1pr*::DVE-1::GFP (plasmid gifted to us by Cole Haynes). pJR18 was injected at 5 ng/μl in N2 animals.

## Staging time course for DD remodeling

Briefly, freshly hatched larvae were transferred to seeded OP50 plates and maintained at 25 °C (timepoint 0). Imaging and analysis of iAChR or synaptic vesicle remodeling was assessed as previously described[23].

## Confocal imaging and analysis

Unless noted otherwise, all strains were immobilized with sodium azide (0.3 M) on a 2% or 5% agarose pad. All confocal images were obtained either using a Yokogawa CSU-10 spinning disk confocal equipped with a 63x objective or a Yokogawa CSU-X1-A1N spinning disk confocal (Perkin Elmer) equipped with EM-CCD camera (Hamamatsu, C9100-50) and 63x oil immersion objective. Analysis of synapse number and fluorescence intensity was performed using FIJI ImageJ software (open source) using defined threshold values acquired from control experiments for each fluorescent marker. Statistical analysis for all synaptic and spine analysis between two groups utilized a student's $t$-test; for analysis where, multiple groups were compared either a one-way or two-way ANOVA was used with the appropriate post hoc test.

**Synaptic analysis.** Background fluorescence was subtracted by calculating the average intensity in a region outside the ROI. All ROIs were 25 μm or 30 μm in length. Quantification of the number of puncta within an ROI had a set threshold of 25-255 and the analyze particles function of ImageJ was used to quantify any particle >4 pixels$^2$. Fluorescence intensity was quantified from the raw integrated fluorescence within the ROI. For quantification of DD neuron synapses, the ROI was defined as either the ventral region anterior to the DD1 soma or the opposing dorsal region. For quantification of the DA/DB neuron synapses the ROI was defined as either the ventral region between DB1 and DB3 or the opposing dorsal region between VB1 and VB3.

**Spine/dendrite analysis.** Spine number was quantified as described previously[27,38]. Briefly, spines were counted within a 30 μm ROI anterior to the soma of DD1. Dendrite length was defined as the anterior extension from DD1 soma to the end of the ventral process.

## EMS screen

The EMS mutagenesis protocol was adapted from ref. [76]. Strain IZ1905 (*flp-13pr*::ACR-12::GFP) was treated with 25 μM ethyl methanesulfonate (EMS, Sigma). After washing, P0 mutagenized animals were recovered. F1 animals were transferred to fresh plates and 8 F2s were isolated per F1 (F2 clonal screen). The F3 progeny of ~400 F2s per round were screened. After 27 rounds of EMS, a total of 3261 haploid genomes were screened. Each isolated candidate mutant was rescreened three times to confirm the phenotype.

## Variant discovery mapping and whole genome sequencing

Mutant strains were backcrossed a single time into the starting strain injected with *unc-122pr*::GFP co-injection marker (IZ2302, enabling the distinction of cross- from self-progeny). F2 animals were isolated onto separate plates and their F3 brood were screened by confocal microscope for synapse elimination. 21 independent homozygous recombinant F2 lines were isolated and pooled together. Worm genomic DNA was prepared for sequencing using Gentra Puregene Tissue Kit DNA purification (Qiagen). Library construction and whole genome sequencing were performed by Novogene. Briefly, NEBNext DNA Library Prep Kit was used for library construction. Pair-end sequencing was performed on Illumina sequencing platform with a read length of 150 bp at each end. Reads were mapped to *C. elegans* reference genome version WS220 and analyzed using the CloudMap pipeline[77] where mismatches were compared to the parental strain as well as to other mutants isolated from the screen[77,78].

## Auxin treatment

NGM control (ethanol) or 4 mM synthetic auxin analog 1 (NAA) (Sigma-Aldrich CAS:317918) plates were made as described[32,33,79]. Plates were seeded with concentrated OP50 and stored at room temperature and kept out of light. Animals were synchronized at hatch and staged onto either control or NAA plates.

## Injection of fluorescent antibodies for in vivo labeling of iAChRs

Mouse monoclonal α-HA antibodies (16B12 Biolegend) coupled to Alexa594 were diluted in injection buffer (20 mM $K_3PO_4$, 3 mM K citrate, 2% PEG 6000, pH 7.5). Antibody was injected into the pseudocoelom of L2/L3 stage wild-type or *dve-1(uf171)* animals as described previously[23,39]. Animals were allowed to recover for six hours on seeded NGM plates. Only animals in which fluorescence was observed in coelomocytes were included in the analysis. A student's $t$-test was used for statistical analysis.

## Photoconversion of Dendra2::RAB-3

Wild-type and *dve-1(uf171)* mutant L1 animals (12–14 h post-hatch) expressing Dendra2::RAB3 were paralyzed for imaging using 1 mM levamisole. Dendra2::RAB-3 puncta in the DA/DB dorsal axonal process were photoconverted using a 405 nm laser at 800 ms for 30 s. Images were acquired immediately following photoconversion and again 10 hours later. Animals were rescued following photoconversion and imaging, and allowed to recover until the subsequent timepoint. Both red and green fluorescent signals were quantified. A student's $t$-test was used for statistical analysis.

## Aldicarb paralysis assays

Aldicarb assays were performed as previously[44]. Strains were scored in parallel, with the researcher blinded to the genotype. Young adult animals (24 h after L4) at room temperature (22–24 °C) were selected (>10 per trial for at least 3 trials) and transferred to a nematode growth medium plate containing 1 mM aldicarb (Sigma-Aldrich CAS:116-06-3). Movement was assessed every 15 min for 2 h. Animals that displayed no movement when prodded (paralyzed) were noted. The percentage of paralyzed animals was calculated at each timepoint.

## Calcium imaging

Transgenic animals expressing *ttr-39pr*::GCaMP6s::SL2::mCherry (GABA neurons) along with *unc-129pr*::Chrimson::SL2::BFP (DA and DB cholinergic neurons) were grown on NGM plates with OP50 containing 2.75 mM All-Trans Retinal (ATR). L4 animals (40 h post-hatch) were staged 24 h prior to experiments on fresh ATR OP50 NGM plates. Imaging was performed using 1-day adults immobilized in hydrogel[27,80]. Animals were transferred to 7.5 μL of hydrogel mix placed on a silanized glass slide and covered with a cover slip. Hydrogel was cured using a handheld UV Transilluminator (312 nm, 3 min). A TTL-controlled 625 nm light guide coupled LED (Mightex Systems) was used for Chrimson photoactivation (-14 mW/cm$^2$). A 556 nm BrightLine single-edge short-pass dichroic beam splitter was positioned in the light path (Semrock)[38]. Data were acquired at 10 Hz for 15 s using Volocity software with 4 × 4 binning. Analysis was performed using ImageJ. DD and VD GABA motor neuron cell bodies were identified by mCherry fluorescence and anatomically identified by position along the ventral nerve cord. Each field typically contained 1–5 GABA motor neurons. Only neurons located anterior to the vulva were included in the analysis. Photobleaching correction was performed on background subtracted images by fitting an exponential function to the data (CorrectBleach plugin, ImageJ). Pre-stimulus baseline fluorescence ($F_0$) was calculated as the average of the corrected background-subtracted data points in the first 4 s of the recording and the corrected fluorescence data were normalized to prestimulus baseline as $\Delta F/F_0$, where $\Delta F = F - F_0$. Peak $\Delta F/F_0$ was determined by fitting a Gaussian function to the $\Delta F/F_0$ time sequence using Multi peak

2.0 (Igor Pro, WaveMetrics). All data collected were analyzed, including failures (no response to stimulation). Peak $\Delta F/F_0$ values were calculated from recordings of >10 animals per genotype. Mean peaks ± SEM were calculated from all peak $\Delta F/F_0$ data values. For all genotypes, control animals grown in the absence of ATR were imaged.

### Single worm tracking
Single worm tracking was carried out on NGM plates seeded with 50 μL of OP50 bacteria, using Worm Tracker 2[81]. Animals were acclimated for 30 s prior to tracking. Worm tracker software version 2.0.3.1, created by Eviatar Yemini and Tadas Jucikas (Schafer lab, MRC, Cambridge, UK), was used to analyze movement[82]. Locomotion paths and movement features were extracted from 5 minutes of continuous locomotion tracking. Scoring of path trajectories was performed blinded to genotype.

### Optogenetic analysis
Behavioral assays were performed with young adults at room temperature (22 °C–24 °C). Animals were grown on plates seeded with OP50 containing 2.7 mM All-Trans Retinal (ATR). Animals were placed on fresh plates seeded with a thin lawn of OP50 containing ATR and allowed to acclimate for 1 min. Dorsal-ventral position was noted prior to recording. Animals were allowed to move freely on plates and recorded with a Mightex X camera for 1 min before stimulus, following this a Mightex LED module was used to stimulate Chrimson (625 nm 14 mW/cm$^2$) continuously for 2 min. Locomotion (trajectory and body bending) was analyzed with WormLab (MBF Bioscience) software. A mid-point bending angle histogram was generated for each animal such that over the span of 2 min body angles were measured and binned by the degree of angle. Depending on the starting position, negative and positive degree angles were assigned dorsal or ventral. Any bending angle >0 but <50° was determined a regular bend. We noted wild-type animals without stimulus rarely make angles >50˚ and qualified any bending angle over 50° as a deep bend. An ANOVA with Dunnett's multiple comparisons test was used for comparisons between pre-stimulus and stimulus in wild type and *dve-1(uf171)*. Student's *t*-test was used when comparing the number of dorsal bends >50° in wild- type vs *dve-1* mutant animals.

### RNAi by feeding
L4 larvae expressing *hsp-6pr::GFP* were cultured with *E. coli* expressing either control double-stranded RNA (empty vector) or targeting *atfs-1* and progeny were allowed to develop to L4 stage at 20 °C. Intestinal GFP fluorescence of L4 stage progeny was measured using the Zeiss Imager M1, 10× objective.

### DVE-1 nuclear localization
DVE-1::GFP was measured in L1 stage DD nuclei of *dve-1(syb1984); ufEx1814(unc-47pr::NLS::mCherry)* animals. ROIs were determined by expression of the nuclear localized mCherry signal. Within the ROI a segmented line was drawn through the nucleus and an intensity profile was created for each nucleus. Fluorescence intensity values for DVE-1::GFP were quantified by averaging the largest 5 intensity values at the peak (roughly 0.5 μm). At least 2 DD nuclei per animal were analyzed. ANOVA with Dunnett's multiple comparison test was used for statistical analysis. For measurements in *unc-30* mutants, an ROI extending posteriorly 30 μm from the base of the pharynx was selected. Red GABA neurons in the head, unaffected by *unc-30* mutation, and the pharynx were used as landmarks. Students *t*-test was used for statistical analysis.

### ChIP-seq data acquisition from ModEncode and de novo motif discovery
modENCODE (www.modencode.org) ChIP-seq data were generated by anti-GFP immunoprecipitation from animals stably expressing DVE-1::GFP (strain OP398). The DVE-1 ChIP-seq dataset included two biological replicates at the late embryo stage as well as control animals. Significant peaks were called using PeakSeq and only peaks that were identified in both biological replicates were considered for analysis. DVE-1::GFP ChIP-seq data and experimental details can be found at http://intermine.modencode.org/release-33/report.do?id=77000654 (DCCid: modENCODE_4804)[83,84]. Peaks were considered mapped to genes if there was at least 80% overlap between the peak maximum read density and a 1 kb region upstream of transcriptional start site using the UCSC table browser intersect function[85]. DVE-1 peaks assigned to promoters were used for de novo motif discovery. The sequences of the complete peak region were retrieved from modENCODE. De novo motifs were identified using the peak-motifs module of RSAT (Regulatory Sequence Analysis Tools)[86,87].

### Pathway analysis
Pathway analysis was performed using both WormCat[56] (http://www.wormcat.com) and WormenrichR[57,88] (https://amp.pharm.mssm.edu/WormEnrichr/). For WormCat analysis, pathways with $p < 0.01$ and Bonferroni FDR < 0.01 were considered enriched. The WormEnrichR pathway enrichment analysis utilized the WikiPathway database[89]. WormEnrichR uses the logarithm of the p-value from a Fischer exact test and multiplying that by the z-score of the expected rank to create a combined score. Pathways with adjusted $p < 0.05$ and combined score >5 were considered enriched (Fig. 7a).

### RNA isolation and RT-qPCR
L1 N2 and *dve-1(uf171)* mutant animals were lysed in 2% SDS, 20% b-ME, 40 mM EDTA, 40 mM Tris-HCl pH 7.5, 2 mg/ml Proteinase K. RNA was isolated (Zymo CAS:R1013) and treated with DNAse (Roche CAS:4716728001). cDNA was prepared using reverse transcriptase (RT) synthesis (Roche CAS:5081955001 and Thermo CAS:SO142). RT-qPCR procedures were followed according to the KAPA SYBR FAST qPCR Kit protocol (CAS:7959397001) and performed with a Bio-Rad CFX96 Real-Time System with a C1000 Touch Thermal Cycler. cDNA was standardized to *act-1*. Primers used are listed in Supplementary Data 3. Representative experiments from at least three repetitions are shown.

### Proteasome and *uba-1* time course and experiments
Worms were hatched synchronously on NGM plates. Wild-type animals were transferred to either 5 μM Bortezomib (MilliporeSignma CAS:179324-69-7) or control (DMSO) plates 10 hours after hatch and allowed to develop until imaging at 24 h after hatch. Wildtype and *uba-1(it129)* mutants were either maintained continuously at 15 °C or transferred from 15 to 25 °C at 10 h after hatch and until imaging at 24 h after hatch.

### Statistics and reproducibility
Summary statistics are included in supplementary data 6.

### Reporting summary
Further information on research design is available in the Nature Portfolio Reporting Summary linked to this article.

## Data availability
All raw data represented in this manuscript is provided in the Supplementary Information/Source Data file. DVE-1::GFP ChIP-seq data and experimental details can be found in the modENCODE project database at http://intermine.modencode.org/release-33/report.do?id=77000654 (DCCid: modENCODE_4804) Source data are provided with this paper.

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

## Acknowledgements

We thank Alexandra Byrne, Dori Schafer, and the members of the Francis lab for critical reading of the manuscript. We also thank Ye Tian and Cole Haynes for generously sharing reagents and Michael Gorczyca and Will Joyce for technical assistance. We would like to thank Mark Alkema and Jeremy Florman for their assistance with single worm tracking and analysis scripts. Some nematode strains were provided by the *Caenorhabditis* Genetics Center which is funded by the NIH National Center for Research Resources, and by the National Bioresource Project Nematode of Japan. This research was supported by NIH RO1NS064263 (M.M.F.), HHMI Gilliam GT11432 (K.A.), and IMSD T32GM135701 (K.A.).

## Author contributions

K.D.A. generated strains, transgenic lines, molecular constructs, confocal microscopy images, and analysis, performed optogenetic behavioral experiments, photoconversion experiments, modencode ChIP-seq analysis, and pathway analysis. S.R. performed all calcium imaging experiments/analysis and conducted single worm tracking. K.B. performed all Bortezomib inhibitor experiments and analysis. C.L. generated most vectors and constructs. J.R. assisted with the generation of CRISPR/Cas9-generated strains. W.A. and M.R. assisted with aldicarb

behavioral assay. C.B. and D.O. assisted with EMS screen and isolation of *dve-1* mutant. C.B. and M.D. aided in CloudMap bioinformatic analysis of the *uf171* mutant. S.L. and A.K.W. performed and analyzed all qPCR experiments. M.M.F. and K.D.A. designed and interpreted results of all experiments and wrote the manuscript.

## Competing interests

The authors declare no competing interests.
