## [Peer Review File · Nature Communications]

The homeodomain transcriptional regulator DVE-1 directs a program for synapse elimination during circuit remodelingREVIEWER COMMENTS

Reviewer #1 (Remarks to the Author):

Reviewer comments- Nature Communications manuscript NCOMMS-22-45948A-Z

Developmental neuronal remodelling is essential for forming functional nervous systems across species. Specifically, elimination of existing connections is a key step in this process. Alexander et al focused on a well-established model of neuronal remodeling- The synaptic remodeling of the dorsal D motor neurons during early *C. elegans* development. During the second larval stage these neurons remodel to place presynaptic boutons on the dorsal side, and relocate postsynaptic terminals to the ventral side for cholinergic input from VA/VB motor neurons. Alexander et al carried out a forward genetic screen and identified a novel regulator of this dorsal synaptic elimination process, the homeodomain transcription factor *dve-1*. Using fluorescent labeling and photoconvertible molecules the authors show that in *dve-1* mutants dorsal synapses persist, yet synaptic growth is not affected. The authors then go on to nicely show that these synapses are functional and demonstrates the importance of this remodeling in dictating the neuronal circuit's function and behavior. The findings regarding the behavioral impact and *dve-1* localization with the regulation of *ulp-4* are exciting.

I have two main issues that could use addressing: the first is that the text is extremely complicated to read and is not very accessible to non *C. elegans* readers. The authors should revise the text to make it as simple as possible. The second major point regards the suggested transcriptional regulation of the UPS by *dve-1*. This result is based on a single experiment using the proteasome inhibitor bortezomib, which does prove that proteasomal function is required for synapse elimination (which has been demonstrated in *C. elegans* by Shen's and Oren-Suissa's labs), but doesn't form a causal link between *dve-1* and transcriptional regulation of any component of the UPS. Thus, the last statement of the results (lines 434-435) is not supported by data, as well as the suggested regulation of the UPS by *dve-1* in figure 6f. This should either be toned down or further experiments are required to support the authors claim. For example, the authors could choose 1-2 UPS targets from table 3, delete the binding site and show that synapse elimination is affected. I think this could strengthen the conclusion of the paper considerably. Minor comments with few suggested ideas:

1. Most figures do not include the exact n number, just $n > 10$. In addition, some figures include the n number inside the graph and some don't. Please make sure the number of animals is stated for all experiments. Along the same lines, in Figure 1G there is an enormous difference between the number of animals analyzed in wild-type vs mutant (11 vs 86). We suggest to randomly pick 11 points out of the 86 and compare them to the mutants. Doing so randomly 1-2 times will ensure a more reliable comparison.

2. The exact nature of the *dev-1(tm4803)* mutation is not shown in Fig 1D (the insertion), and only described in the text. The information about the mutation and the described structure of the protein (helices I, II, III) could be included alongside the evolutionary conservation.

3. Line 107- no experiments to show autonomy have actually been done, such GABAergic-specific rescue of *dve-1* of the synapse maintenance phenotype.

4. Lines 130-137. Order in which *ced-3* results are presented is confusing. I find it more logical to first discuss the negative result of *ced-3* effect on DD dorsal synapse elimination and only then present the (interesting) result of the partial removal of the ventral Rab-3 clusters.

5. Figure 1E and S1.3A, B: A time course of mCherry::Rab3 (SV) in *dve-1* mutants, could be an important addition to further solidify the claim that there is no effect of *dve-1* on pre-synaptic formation on the ventral side.

6. Why don't the authors show statistical analysis for the synapse visualization in Figure 2B?

7. Lines 206-207 – "Prior to photoconversion, clusters of green Dendra2::RAB-3 fluorescence were distributed along the length of cholinergic axons in the dorsal nerve cord"- please add an image showing this result.

8. The photoconversion experiments are one example of where the text could be simplified. A side by side comparison with supplementary fig. S2.1A-E is needed. Moving S2.1A-E to Fig 2 will help the reader digest the data better. In addition, the authors should include an image of the Dendra RFP and GFP channels merged together.

9. Figure 3A: the texts mention the percentage of DD responding neurons in wild type and *dve-1* mutants, but there is no data to back it up (heat maps of individual worms + graph of traces is needed).

10. The ModENCODE data in file S5 does include a few interesting candidates for the synaptic targets of the UPS. The most obvious one is UNC-40, shown by Salzberg et al to be targeted for degradation by the UPS via the E3 ligase *sel-10*, resulting in sex-specific synapse pruning. Jean-Louis Bessereau's lab has shown in multiple papers that UNC-40 organizes NMJ GABA synapses through the MADD-4 ligand. Even though this manuscript doesn't address the synaptic targets of the UPS, but focuses on events upstream to UPS, I think this should be addressed in the discussion.

11. Line 290 – there is not figure for animals that lack chrimson expression – perhaps remove from text.

12. Line 296- referral to figure is missing.

13. Figure 5 – the quantification in F is missing the K327R group.

14. Line 324: (Figure 4F, G) ◊ there is no fig. G...

15. Line 436: there is no Fig. 6G, do the authors mean Fig 6F.

16. Line 482: same as in line 436.

Reviewer #2 (Remarks to the Author):

Review_NatureComm

In this study, the authors identified DVE-1 homeobox protein as a novel regulator of synaptic remodeling. *C. elegans* GABAergic motor neurons undergo synaptic remodeling during larval development, where dorsal and ventral processes 'exchange' their fates by relocating the postsynaptic sites from the dorsal process to the ventral process, while the presynaptic sites are relocated from the ventral processes to the dorsal process. From a forward genetic screening of the abnormal dorsal localization of the postsynaptic marker (ACR-12::GFP) they developed previously, the authors isolated a missense mutant of *dve-1*. Interestingly, they found that *dev-1* mutants have no significant defects in the formation of postsynaptic sites in the ventral processes or the remodeling of the presynaptic sites. This suggests that the function of *dve-1* is specifically required for the elimination of the dorsal postsynaptic sites during remodeling. The authors then showed that ectopic postsynaptic sites of DDs in the dorsal process are opposed by the cholinergic presynaptic marker, RAB-3, from the dorsal cholinergic motor neurons (DA/DB), suggesting ectopic synaptic connections in the *dve-1* mutant. The authors used optogenetics and a single worm tracking system to demonstrate that these ectopic synaptic connections in *dve-1* mutant cause functional defects at the circuit and behavioral level. The endogenously expressed DVE-1::GFP is observed in the nuclei of DD neurons, whose expression is dependent on UNC-30 which determined the cell fate of these neurons. The nuclear localization of DVE-1 requires its deSUMOylation by ULP-4, as previously shown in the intestinal cells. The authors observed a synapse elimination defect in the *ulp-4* mutants similar to *dve-1* mutant, suggesting that the nuclear localization of DVE-1 is critical for its function in synapse elimination. Lastly, the authors examined available ChIP sequence data and found that genes required for the mitochondrial UPR pathway and the ubiquitin-proteasome system as potential targets of DVE-1. Genetic and pharmacological perturbations of these pathways revealed that UPS is a potential mechanism by

which DVE-1 eliminates postsynaptic sites in DD neurons.

I found this work very interesting and important in better understanding the mechanisms that underlie developmental synaptic remodeling for several reasons.

First, this is the first to show the mechanism of postsynaptic remodeling in the GABAergic motor neurons in *C. elegans*. Remodeling of the presynaptic sites of DD neurons is relatively well studied, thanks to the availability of the various presynaptic markers. The author's group previously showed that ACR-12::GFP is localized at the postsynaptic dendritic spine-like protrusions of DD neurons, and can be used as a specific postsynaptic marker. This allowed them to conduct a visual-based forward genetic screening to isolate mutants with defective postsynaptic remodeling and successfully identified *dve-1* as a novel regulator of postsynaptic remodeling.

Second, the authors showed that the *dve-1* mutant had a significant defect in eliminating dorsal postsynaptic sites, without apparent defects in other processes of DD remodeling, such as the formation of dorsal presynaptic sites and ventral postsynaptic sites. This observation strongly suggests that the remodeling of DD neurons is not a simple consequence of axodendritic polarity reversal, but rather a combination of individual events. This is consistent with the previous observation by Kurup et al., 2015 showing that there is no MT polarity change during DD remodeling.

Third, the authors showed that *dve-1* mutant worms exhibit behavioral defects that are likely to be caused by the ectopic connections between DDs and DAs/DBs, demonstrating the physiological significance of developmental synaptic remodeling.

Lastly, the authors found the ubiquitin-proteasome system as a potential mechanism of *dve-1*-mediated postsynapse elimination.

The authors' major conclusions are well supported by a wide variety of genetic, molecular and imaging tools, and this work has significant impacts on advancing our understanding of the genetic basis of neuronal remodeling, which should be of general interest to broad readers of *Nature Communications*, particularly those in the neuroscience and developmental biology fields. I would like the authors to incorporate the following points in their manuscript before it is published in *Nature Communications*.

Major comments:

1. It was unexpected (to me) to see enlarged RAB-3 puncta in the cholinergic motor neurons without affecting the size and number of active zone in the *dve-1* mutant. Are all ectopic dorsal ACR-12::GFP puncta in DD neurons of *dve-1* mutant apposed to UNC-10 (or other active zone markers) in DA/DB neurons? Please discuss potential mechanisms of how ectopic postsynaptic sites in DDs cause an enlarged SV pool in DA/DB.
2. If DA/DB have increased synaptic strength/activity, do *dve-1* mutants exhibit hypersensitivity to aldicarb?
3. *dve-1* mutant animals tend to bend dorsally which cannot be explained directly by the ectopic synapse connection from DA/DB to DDs as it should result in the dorsal relaxation to cause more ventral bending. The authors argue that increased acetylcholine release onto dorsal muscles of *dve-1*. First, it is helpful for readers if the authors explain that DA/DB form dyadic synapses with dorsal muscles and VD dendrite, and the authors predict that DA/DB also form dyadic synapses with muscles and ectopic dorsal postsynapses of DDs in *dve-1* mutant. Second, the alternative explanation of this dorsal bending behavior of *dve-1* could be explained by a defect in DD GABAergic presynaptic activity due to the aberrant synaptic input from DA/DB to the dorsal DD neurites. The authors may test this hypothesis by silencing DD neurons (for example by expressing *HisCl1*) in *dve-1* mutant to test the enhancement of dorsal bending preferences. If this is technically difficult, I would like to see alternative models to be discussed in the discussion.
4. *Dve-1* mutant phenotype is epistatic to the precautionary remodeling phenotype of the *oig-1* mutant. The authors argue that these genes function in parallel pathways to control DD remodeling. The alternative is that *dve-1* functions downstream of *oig-1* for the elimination of DD dorsal postsynaptic sites. Have the authors tested if *dve-1* expression or nuclear localization depends on *oig-1*?
5. The authors found that *dve-1* expression depends on *unc-30*, and found putative *unc-30*-binding sites in the *dve-1* promoter. Can the authors mutate *unc-30* binding sites in the *dve-1* promoter to test their hypothesis of direct regulation of *dve-1* expression by *unc-30*?
6. *Dve-1* expression persists after DD remodeling completes. During larval development, each DD neuron increases the number of postsynaptic sites as the worm grows. Does this mean that *dve-1* is continuously required to eliminate synapses in the dorsal process after remodeling? This can be

tested either by late knockdown of *dve-1* using the auxin degron system or by late rescue of *dve-1* using the heat-shock promoter.

7. The authors showed that *ulp-4* is required for the nuclear localization of VDE-1, and *ulp-4* mutant exhibits a failure of postsynapse removal in the dorsal process of DD. As GABAergic-neuron-specific expression of *ulp-4* rescues the synapse remodeling phenotype, the authors concluded that nuclear localization of DVE-1 in DD neurons is essential for the elimination of dorsal postsynaptic sites. The authors should directly test the cell-autonomous role of *dve-1* in synapse elimination in DD neurons by rescuing the synaptic and behavioral defects using transgenes expressing *dve-1* cDNA specifically in DD neurons.

8. The authors showed that pharmacological perturbation of the ubiquitin-proteasome system results in defects in postsynapse elimination. The authors have several candidate UPS factors whose expression might be controlled by DVE-1. Have the authors tested the mutants of some of these candidate genes in postsynapse elimination? Also, the authors should test the temperature-sensitive mutant of *uba-1* which is widely used to disrupt the UPS.

9. Have the authors examined if *myrf-1* and *myrf-2* mutants have postsynaptic remodeling defects in DD neurons?

Minor comments

1. In Figure 5, Please include a zoomed-out image that covers part of the ventral nerve cord to show DVE-1::GFP expression is exclusively in DD neurons.

2. Tables 1-3, some of the genotypes have low sample sizes. Please increase the sample size to a minimum of 15.

3. Lines 171-172 and 192-193, please add a few words of explanation for 'Using an established approach' and 'Using a previously developed strategy', to help readers better understand what these are. For example, 'for cell-specific labeling using split-GFP'...

Reviewer #3 (Remarks to the Author):

This is an impressive study uncovering the function of the conserved DVE-1/SATB transcription factor in synapse elimination during circuit remodeling in *C. elegans*.

The authors leverage the strengths of *C. elegans* and the well-known phenomenon of DD GABAergic neuron remodeling to provide key insights on the genetic programs involved in synapse elimination during development. The paper is well-written, and conclusions are generally supported by the data. Key conclusions are:

- a). Remodeling mechanisms of pre- and post-synaptic sites in DD neurons can be genetically separated, arguing for distinct mechanisms controlling pre- and post-synaptic site remodeling.
- b). Forward genetic screen and extensive synaptic analysis establish a selective requirement for DVE-1/SATB in DD synapse elimination (no effect on formation of new postsynaptic sites during development).
- c). DVE-1 is a known regulator of mitochondrial stress response, but the experiments here suggest that this previous function is not relevant for DD remodeling.
- d). Elegant photoconversion experiments show increased stability of cholinergic presynaptic vesicles in the absence of *dve-1*.
- e). Automated worm tracking and photostimulation experiments describe a robust effect on locomotion of *dve-1* mutants.
- f). DVE-1 removes OIG-1 antagonism, significantly extending previous findings on *oig-1*.
- g). Localization of DVE-1 to DD nuclei is essential for synapse elimination.
- h). Analysis of putative DVE-1 target genes and pharmacology implicate the ubiquitin-proteasome system in synapse elimination.

Major comments:

Line 369: "DVE-1 transcriptional targets" can change to "putative direct target genes of DVE-1". Such change is needed because a TF binding event on the genome does not always have an effect on gene transcription. The authors also make great use of available ChIP-Seq data, but more

details are needed. What developmental stage was used for DVE-1 ChIP-Seq? If a fosmid-based reporter was used, then the overexpression of DEV-1 could affect, for example, its binding affinity, and this limitation should be acknowledged. Lastly, given that the binding preference (motif) of DVE-1 is not known, can the authors use the coordinates of DVE-1 ChIP-Seq peaks and conduct de novo motif analysis? Even a simple analysis of genomic distribution of the peaks (e.g., promoter, intron, etc) could be informative.

The data suggest very intriguing non-cell autonomous effects: cholinergic presynaptic sites onto DD neurons persist into adulthood in *dve-1* mutants, and *dve-1* promotes destabilization of vesicle assemblies in cholinergic neurons. First, these appear to be two distinct non-cell autonomous phenotypes, and the authors should clarify this in the text. Second, at a mechanistic level, can the authors discuss what triggers these non-cell autonomous effects (e.g., a secreted factor under DVE-1 transcriptional control)?

The paper states that DVE-1 is implicated in the transcriptional control of the ubiquitin-proteasome system in synapse elimination. Although there is evidence (e.g., GO analysis on DEV-1 targets, proteasome inhibition experiments) supporting this possibility, the authors have yet to formally establish a transcriptional link, i.e., to identify which direct targets of DVE-1 are important for proteasome function and synaptic remodeling. Moreover, the proteasome inhibitor (bortezomid) can have broad effects on degradation of various proteins (because high dosage leads to larval arrest), and this could confound data interpretation. For example, can the authors test whether bortezomid has a specific effect on the iAChR phenotype and/or cholinergic presynaptic sites, or does it also affect degradation of other proteins that have no role on DD synaptic remodeling? In other words, how "healthy or functional" are the DD neurons after bortezomid treatment?

Minor comments:

- Because DA and DB form dyadic synapses onto GABA and dorsal muscle, the synaptic analysis of *dve-1* can be strengthened by using a marker for ACh receptors (e.g., UNC-38) expressed in dorsal muscle. Along the same lines, the surprising result on the active zone marker UNC-10 can be corroborated with a second marker, such as SYD-1.

- In Abstract: "DVE-1 targets" should be "putative DVE-1 target genes"

- Figure 1D illustrates that DVE-1 contains two homeodomains with similarity to the HOX homeodomain. In *C. elegans*, only 6 of the ~100 homeodomain transcription factors belong to the HOX family, and DVE-1 is not a HOX gene. Because there is huge confusion in the literature that every homeodomain TF is a Hox gene (which is not true), I would suggest to simply indicate in Fig 1D that DVE-1 has two homeodomains, essentially to change "HOX" to "HD" in the figure, as previous SATB studies have done:

<https://doi.org/10.1093/nar/gkm1151>

<https://pubmed.ncbi.nlm.nih.gov/20351170/#&gid=article-figures&pid=fig-1-uid-0>

- Line 157: was the wild-type *dve-1* gene expressed under a GABA-specific promoter? It is important to state this clearly in text, as the paper argues for a cell-autonomous DVE-1 function.

- Can the authors rule out the possibility that *ulp-4* affects the transcription of *dve-1*?

- Line 282: 6G should be 6F.

- The relevance of SATB to human syndromes could be mentioned earlier (e.g., Abstract, Intro).

Response to Reviewers

We thank the reviewers for their helpful comments and suggestions. We address each of their specific points below. We hope our responses and revisions have satisfactorily clarified the issues raised.

Reviewer #1 (Remarks to the Author):

Developmental neuronal remodeling is essential for forming functional nervous systems across species. Specifically, elimination of existing connections is a key step in this process. Alexander et al focused on a well-established model of neuronal remodeling- The synaptic remodeling of the dorsal D motor neurons during early *C. elegans* development. During the second larval stage these neurons remodel to place presynaptic boutons on the dorsal side and relocate postsynaptic terminals to the ventral side for cholinergic input from VA/VB motor neurons. Alexander et al carried out a forward genetic screen and identified a novel regulator of this dorsal synaptic elimination process, the homeodomain transcription factor *dve-1*. Using fluorescent labeling and photoconvertible molecules the authors show that in *dve-1* mutants dorsal synapses persist, yet synaptic growth is not affected. The authors then go on to nicely show that these synapses are functional and demonstrates the importance of this remodeling in dictating the neuronal circuit's function and behavior. The findings regarding the behavioral impact and *dve-1* localization with the regulation of *ulp-4* are exciting.

We thank the reviewer for their positive assessment.

I have two main issues that could use addressing:

First is that the text is extremely complicated to read and is not very accessible to non-*C. elegans* readers. The authors should revise the text to make it as simple as possible.

We have revised the text considerably for clarity and simplicity.

The second major point regards the suggested transcriptional regulation of the UPS by *dve-1*. This result is based on a single experiment using the proteasome inhibitor bortezomib, which does prove that proteasomal function is required for synapse elimination (which has been demonstrated in *C. elegans* by Shen's and Oren-Suissa's labs) but doesn't form a causal link between *dve-1* and transcriptional regulation of any component of the UPS. Thus, the last statement of the results (lines 434-435) is not supported by data, as well as the suggested regulation of the UPS by *dve-1* in figure 6f.

This should either be toned down or further experiments are required to support the authors claim. For example, the authors could choose 1-2 UPS targets from table 3, delete the binding site and show that synapse elimination is affected. I think this could strengthen the conclusion of the paper considerably.

We thank the reviewer pointing this out. We performed several additional experiments to address this issue; however, it is challenging because of the potential for redundancy in the UPS pathway. For example, our analysis revealed 5 different E3 ligases as potential targets of DVE-1 transcriptional

regulation. We first analyzed the expression of four putative DVE-1 targets identified from our ChIP-seq analysis by qPCR. We found that expression of two of these (*cul-5* and *spat-3*) is significantly downregulated in *dve-1* mutants while *eel-1* appears upregulated (Figure S7.2E). We also investigated remodeling using null or loss-of-function alleles of several candidate genes implicated from our ChIP-seq analysis (E3s *cul-5*, *eel-1*, *spat-3* and associated UPS genes *try-6*, *cpi-2*, *spsb-1*), but found that synapse elimination was unaffected in single mutants carrying these alleles. We therefore tested a temperature sensitive allele of the sole E1 homolog, *uba-1*. We found that synapse elimination was significantly delayed in *uba-1* mutants shifted to the restrictive temperature (Figure 7E,F). Nonetheless, we acknowledge the possibility that additional DVE-1-regulated pathways may also contribute and we have modified the text to include this point and better reflect the limitations of our analysis. We show the new qPCR and *uba-1* results but, for simplicity, do not currently include the negative data for the single mutants in the revised manuscript. We can of course include if the reviewer feels they add to the paper.

Minor comments with few suggested ideas:

1. Most figures do not include the exact n number, just n >10. In addition, some figures include the n number inside the graph and some don't. Please make sure the number of animals is stated for all experiments.

We now provide the exact n for each figure.

Along the same lines, in Figure 1G there is an enormous difference between the number of animals analyzed in wild-type vs mutant (11 vs 86). We suggest to randomly pick 11 points out of the 86 and compare them to the mutants. Doing so randomly 1-2 times will ensure a more reliable comparison. We increased the n for this figure (now Figure 2H). Similar numbers of wild type and *dve-1(uf171)* mutants are now included in the analysis.

2. The exact nature of the *dve-1(tm4803)* mutation is not shown in Fig 1D (the insertion), and only described in the text. The information about the mutation and the described structure of the protein (helices I, II, III) could be included alongside the evolutionary conservation.

We now indicate (now in Figure 1C) the positions of the insertion and deletion for *dve-1(tm4803)* and the predicted locations of the helices (I,II, and III) within homedomain I.

3. Line 107- no experiments to show autonomy have actually been done, such GABAergic-specific rescue of *dve-1* of the synapse maintenance phenotype.

We now include data in Figure 1D,E showing that GABA-specific expression of *dve-1* cDNA is sufficient for rescue.

4. Lines 130-137. Order in which *ced-3* results are presented is confusing. I find it more logical to first discuss the negative result of *ced-3* effect on DD dorsal synapse elimination and only then present the (interesting) result of the partial removal of the ventral Rab-3 clusters.

The effects of *ced-3* mutation on RAB-3 clusters have been reported previously (Meng et. al. 2015). We included it here as confirmatory. We have reordered the text as suggested (lines 136-142).

5. Figure 1E and S1.3A, B: A time course of mCherry::Rab3 (SV) in *dve-1* mutants, could be an important addition to further solidify the claim that there is no effect of *dve-1* on pre-synaptic formation on the ventral side.

We thank the reviewer for this suggestion. We now include new data showing the timecourse of mCherry::RAB-3 (SVs) remodeling in *dve-1* mutants (Figure S2.1C). SV clusters begin to appear on the dorsal side by 14 hrs post-hatch and are completely removed from the ventral side by 24 hrs post-hatch in *dve-1(uf171)* mutants, similar to wild type, further demonstrating that DVE-1 is not required for the formation of dorsal presynaptic sites.

6. Why don't the authors show statistical analysis for the synapse visualization in Figure 2B?

Statistical comparison (student's t-test) of the percentage of apposed ACR-12::GFP (AChR) clusters to mCherry::RAB-3 (SV) clusters versus the shifted controls is now included in Figure 3D of the revised manuscript. We also include in this figure new analysis of AChR localization relative to the active zone marker ELKS-1 in cholinergic axons (Figure 3C).

7. Lines 206-207 – “Prior to photoconversion, clusters of green Dendra2::RAB-3 fluorescence were distributed along the length of cholinergic axons in the dorsal nerve cord”- please add an image showing this result.

We now include an image of wild type and *dve-1(uf171)* mutant Dendra2::RAB-3 prior to photoconversion in Figure S3.1A.

8. The photoconversion experiments are one example of where the text could be simplified. A side by side comparison with supplementary fig. S2.1A-E is needed. Moving S2.1A-E to Fig 2 will help the reader

digest the data better. In addition, the authors should include an image of the Dendra RFP and GFP channels merged together.

We thank the reviewer for pointing this out. As suggested, we have significantly reorganized the presentation of these experiments both in the text and in Figures 3 and S3.1. We now show the merged red and green channels in Figure 3G,I. We also include a side-by-side comparison of the red and green channels in Figure S3.1B-F to make the data easier to follow.

9. Figure 3A: the texts mention the percentage of DD responding neurons in wild type and *dve-1* mutants, but there is no data to back it up (heat maps of individual worms + graph of traces is needed). We measured the calcium responses of individual motor neurons. Unfortunately, these data are not effectively represented in a heat map. We show a scatter plot of all peak responses in Figure 4A. All recordings where we did not measure a significant fluorescence increase following photostimulation were recorded as zero values and considered non-responders. We have now clarified this point in the methods (lines 727-730).

10. The ModENCODE data in file S5 does include a few interesting candidates for the synaptic targets of the UPS. The most obvious one is UNC-40, shown by Salzberg et al to be targeted for degradation by the UPS via the E3 ligase *sel-10*, resulting in sex-specific synapse pruning. Jean-Louis Bessereau's lab has shown in multiple papers that UNC-40 organizes NMJ GABA synapses through the MADD-4 ligand. Even though this manuscript doesn't address the synaptic targets of the UPS, but focuses on events upstream to UPS, I think this should be addressed in the discussion.

Good point. We now include discussion of this point in the Discussion (lines 544-548). Though not included in the manuscript, we also note that mutation of the E3 ligase *sel-10* has no effect on synapse elimination in DD neurons, suggesting an alternate mechanism.

11. Line 290 – there is not figure for animals that lack chromosom expression – perhaps remove from text. We thank the reviewer for pointing this out. Removed from the text.

12. Line 296- referral to figure is missing.

This line has been removed from the final text.

13. Figure 5 – the quantification in F is missing the K327R group.

Now added in Figure 6F.

14. Line 324: (Figure 4F,G) \diamond there is no fig. G..., Line 436: there is no Fig. 6G, do the authors mean Fig 6F.

Line 482: same as in line 436.

We thank the reviewer for pointing this out. Now corrected or deleted (lines 348, 465 of the revised manuscript).

Reviewer #2 (Remarks to the Author)

In this study, the authors identified DVE-1 homeobox protein as a novel regulator of synaptic remodeling. *C. elegans* GABAergic motor neurons undergo synaptic remodeling during larval development, where dorsal and ventral processes 'exchange' their fates by relocating the postsynaptic sites from the dorsal process to the ventral process, while the presynaptic sites are relocated from the ventral processes to the dorsal process. From a forward genetic screening of the abnormal dorsal localization of the postsynaptic marker (ACR-12::GFP) they developed previously, the authors isolated a missense mutant of *dve-1*. Interestingly, they found that *dev-1* mutants have no significant defects in the formation of postsynaptic sites in the ventral processes or the remodeling of the presynaptic sites. This suggests that the function of *dve-1* is specifically required for the elimination of the dorsal postsynaptic sites during remodeling. The authors then showed that ectopic postsynaptic sites of DDs in the dorsal process are opposed by the cholinergic presynaptic marker, RAB-3, from the dorsal cholinergic motor neurons (DA/DB), suggesting ectopic synaptic connections in the *dve-1* mutant. The authors used optogenetics and a single worm tracking system to demonstrate that these ectopic synaptic connections in *dve-1* mutant cause functional defects at the circuit and behavioral level. The endogenously expressed DVE-1::GFP is observed in the nuclei of DD neurons, whose expression is dependent on UNC-30 which determined the cell fate of these neurons. The nuclear localization of DVE-1 requires its deSUMOylation by ULP-4, as previously shown in the intestinal cells. The authors observed a synapse elimination defect in the *ulp-4* mutants similar to *dve-1* mutant, suggesting that the nuclear localization of DVE-1 is critical for its function in synapse elimination. Lastly, the authors examined available ChIP sequence data and found that genes required for the mitochondrial UPR pathway and the ubiquitin-proteasome system as potential targets of DVE-1. Genetic and pharmacological perturbations of these pathways revealed that UPS is a potential mechanism by which DVE-1 eliminates postsynaptic sites in DD neurons. I found this work very interesting and important in better understanding the mechanisms that underlie developmental synaptic remodeling for several reasons.

First, this is the first to show the mechanism of postsynaptic remodeling in the GABAergic motor neurons in *C. elegans*. Remodeling of the presynaptic sites of DD neurons is relatively well studied, thanks to the availability of the various presynaptic markers. The author's group previously showed that ACR-12::GFP is localized at the postsynaptic dendritic spine-like protrusions of DD neurons, and can be used as a specific postsynaptic marker. This allowed them to conduct a visual-based forward genetic screening to isolate mutants with defective postsynaptic remodeling and successfully identified *dve-1* as a novel regulator of postsynaptic remodeling.

Second, the authors showed that the *dve-1* mutant had a significant defect in eliminating dorsal postsynaptic sites, without apparent defects in other processes of DD remodeling, such as the formation of dorsal presynaptic sites and ventral postsynaptic sites. This observation strongly suggests that the remodeling of DD neurons is not a simple consequence of axodendritic polarity reversal, but rather a combination of individual events. This is consistent with the previous observation by Kurup et al., 2015 showing that there is no MT polarity change during DD remodeling.

Third, the authors showed that *dve-1* mutant worms exhibit behavioral defects that are likely to be caused by the ectopic connections between DDs and DAs/DBs, demonstrating the physiological significance of developmental synaptic remodeling.

Lastly, the authors found the ubiquitin-proteasome system as a potential mechanism of *dve-1*-mediated postsynapse elimination.

The authors' major conclusions are well supported by a wide variety of genetic, molecular and imaging tools, and this work has significant impacts on advancing our understanding of the genetic basis of neuronal remodeling, which should be of general interest to broad readers of *Nature Communications*, particularly those in the neuroscience and developmental biology fields. I would like the authors to incorporate the following points in their manuscript before it is published in *Nature Communications*.

We thank the reviewer for their positive assessment.

Major comments:

1. It was unexpected (to me) to see enlarged RAB-3 puncta in the cholinergic motor neurons without affecting the size and number of active zone in the *dve-1* mutant. Are all ectopic dorsal ACR-12::GFP puncta in DD neurons of *dve-1* mutant apposed to UNC-10 (or other active zone markers) in DA/DB neurons?

We did several experiments to follow up on this and now include new data. We found an increase in a second synaptic vesicle marker (RAB-3) in cholinergic axons (Figure S3.1H), similar to our previous findings for the SNB-1 synaptic vesicle marker (Figure 3K). We also determined the localization of ACR-12::GFP clusters relative to the cholinergic active zone marker ELKS-1 (Figure 3C). The majority of ACR-12::GFP clusters are apposed to active zones in cholinergic axons of *dve-1* mutants. Further, similar to our prior analysis of UNC-10/RIM, we did not observe changes in the size or number of ELKS-1 puncta in *dve-1* mutants (Figure S3.1J,K).

Please discuss potential mechanisms of how ectopic postsynaptic sites in DDs cause an enlarged SV pool in DA/DB.

This is an interesting question. We speculate that the retention of juvenile synaptic vesicle clusters in *dve-1* mutants occurs in parallel with the formation of new synaptic connections between cholinergic DA/B motor neurons and post-embryonic born VD GABAergic neurons. Our data suggesting that the synaptic vesicle pool is expanded at cholinergic synaptic terminals is consistent with the idea that the retention of juvenile SV assemblies and the addition of new SV assemblies during maturation of the circuit is additive, leading to an enlarged pool. We now include discussion of this point (lines 525-531).

2. If DA/DB have increased synaptic strength/activity, do *dve-1* mutants exhibit hypersensitivity to aldicarb?

We show that *dve-1* mutants are hypersensitive to aldicarb in Figure S4.1B.

3. *dve-1* mutant animals tend to bend dorsally which cannot be explained directly by the ectopic synapse connection from DA/DB to DDs as it should result in the dorsal relaxation to cause more ventral bending. The authors argue that increased acetylcholine release onto dorsal muscles of *dve-1*. First, it is helpful for readers if the authors explain that DA/DB form dyadic synapses with dorsal muscles and VD

dendrite, and the authors predict that DA/DB also form dyadic synapses with muscles and ectopic dorsal postsynapses of DDs in *dve-1* mutant.

Good point. We now describe the connectivity of DA/B in the text on page 15 (lines 297-301).

Second, the alternative explanation of this dorsal bending behavior of *dve-1* could be explained by a defect in DD GABAergic presynaptic activity due to the aberrant synaptic input from DA/DB to the dorsal DD neurites. The authors may test this hypothesis by silencing DD neurons (for example by expressing *HisCl1*) in *dve-1* mutant to test the enhancement of dorsal bending preferences. If this is technically difficult, I would like to see alternative models to be discussed in the discussion.

We sought to test the functionality of DD inhibitory synapses by measuring behavioral responses to DD photostimulation in animals expressing *Chrimson* under control of the *flp-13* promoter. Though we did not observe significant differences between wild type and *dve-1* mutants in these experiments, the expression of *Pf13::Chrimson* in other neurons additional to DD neurons complicates interpretation of this result (and would also present an issue for silencing experiments based on expression using the *flp-13* promoter). Acknowledging this limitation, we have modified the text to include the possibility of alternate models (lines 327-328).

4. *Dve-1* mutant phenotype is epistatic to the precocious remodeling phenotype of the *oig-1* mutant. The authors argue that these genes function in parallel pathways to control DD remodeling. The alternative is that *dve-1* functions downstream of *oig-1* for the elimination of DD dorsal postsynaptic sites. Have the authors tested if *dve-1* expression or nuclear localization depends on *oig-1*?

We thank the authors for this suggestion. We now include experiments showing that *pdve-1::DVE-1::GFP* expression is not altered in *oig-1* mutants (Figure 5A).

5. The authors found that *dve-1* expression depends on *unc-30*, and found putative *unc-30*-binding sites in the *dve-1* promoter. Can the authors mutate *unc-30* binding sites in the *dve-1* promoter to test their hypothesis of direct regulation of *dve-1* expression by *unc-30*?

We thank the authors for this suggestion. We deleted putative *UNC-30* binding sites from the *dve-1* promoter region of a rescuing *Pdve-1::DVE-1::gpf* transgene (Haynes, 2007) and found a significant decrease of *DVE-1::GFP* fluorescence in GABA neurons but not intestinal cells (Figure S6.1E), consistent with our model.

6. Dve-1 expression persists after DD remodeling completes. During larval development, each DD neuron increases the number of postsynaptic sites as the worm grows. Does this mean that *dve-1* is continuously required to eliminate synapses in the dorsal process after remodeling? This can be tested either by late knockdown of *dve-1* using the auxin degron system or by late rescue of *dve-1* using the heat-shock promoter.

As suggested by the reviewer we depleted neuronal DVE-1 at various developmental time points using the auxin inducible degron (AID) system. Either continuous depletion of DVE-1 for 50 hrs or time-restricted depletion prior to/during remodeling disrupted the elimination of dorsal juvenile synapses. In contrast, we did not observe ectopic dorsal AChR clusters when DVE-1 was depleted after remodeling. These results argue against a continuous requirement for DVE-1 to support ongoing removal of dorsal synapses in the mature circuit. These new results are presented in Figure 2A-D of the revised manuscript.

7. The authors showed that *ulp-4* is required for the nuclear localization of DVE-1, and *ulp-4* mutant exhibits a failure of postsynapse removal in the dorsal process of DD. As GABAergic-neuron-specific expression of *ulp-4* rescues the synapse remodeling phenotype, the authors concluded that nuclear localization of DVE-1 in DD neurons is essential for the elimination of dorsal postsynaptic sites. The authors should directly test the cell-autonomous role of *dve-1* in synapse elimination in DD neurons by rescuing the synaptic and behavioral defects using transgenes expressing *dve-1* cDNA specifically in DD neurons.

Thank you for this suggestion. We now show in Figure 1D-E that GABA-specific expression of the wild-type *dve-1* cDNA is sufficient for rescue.

8. The authors showed that pharmacological perturbation of the ubiquitin-proteasome system results in defects in postsynapse elimination. The authors have several candidate UPS factors whose expression might be controlled by DVE-1. Have the authors tested the mutants of some of these candidate genes in postsynapse elimination? Also, the authors should test the temperature-sensitive mutant of *uba-1* which is widely used to disrupt the UPS.

See also response to Reviewer 1. We investigated synapse elimination in several strains carrying single gene mutations in candidates implicated from our ChIP-seq analysis (E3s *cul-5*, *eel-1*, *spat-3* and associated UPS genes *try-6*, *cpi-2*, *spsb-1*); however, we did not observe significant effects on synapse

elimination in these single mutant strains, perhaps to due to the high level of redundancy in the ubiquitin system. As suggested by the reviewer, we also tested the temperature sensitive allele of the sole E1 homolog, *uba-1*, and found that synapse elimination was significantly delayed at the restrictive temperature (Figure 7E,F), supporting the importance of UPS in synapse elimination. We propose that DVE-1 transcriptional regulation of UPS genes is important for synapse elimination; however, we acknowledge additional DVE-1-regulated pathways may also contribute.

9. Have the authors examined if *myrf-1* and *myrf-2* mutants have postsynaptic remodeling defects in DD neurons?

We have examined *myrf-2* mutants but did not observe a significant alteration in the remodeling of AChRs. We have not included this piece of negative data in the manuscript.

Minor comments

1. In Figure 5, Please include a zoomed-out image that covers part of the ventral nerve cord to show DVE-1::GFP expression is exclusively in DD neurons.

Now included in Figure S6.1A.

2. Tables 1-3, some of the genotypes have low sample sizes. Please increase the sample size to a minimum of 15.

We have increased the sample sizes.

3. Lines 171-172 and 192-193, please add a few words of explanation for 'Using an established approach' and 'Using a previously developed strategy', to help readers better understand what these are. For example, 'for cell-specific labeling using split-GFP'...

Done.

Reviewer #3 (Remarks to the Author):

This is an impressive study uncovering the function of the conserved DVE-1/SATB transcription factor in synapse elimination during circuit remodeling in *C. elegans*. The authors leverage the strengths of *C. elegans* and the well-known phenomenon of DD GABAergic neuron remodeling to provide key insights on the genetic programs involved in synapse elimination during development. The paper is well-written, and conclusions are generally supported by the data.

Key conclusions are:

- a). Remodeling mechanisms of pre- and post-synaptic sites in DD neurons can be genetically separated, arguing for distinct mechanisms controlling pre- and post-synaptic site remodeling.
- b). Forward genetic screen and extensive synaptic analysis establish a selective requirement for DVE-1/SATB in DD synapse elimination (no effect on formation of new postsynaptic sites during development).
- c). DVE-1 is a known regulator of mitochondrial stress response, but the experiments here suggest that this previous function is not relevant for DD remodeling.
- d). Elegant photoconversion experiments show increased stability of cholinergic presynaptic vesicles in the absence of *dve-1*.
- e). Automated worm tracking and photostimulation experiments describe a robust effect on locomotion of *dve-1* mutants.
- f). DVE-1 removes OIG-1 antagonism, significantly extending previous findings on *oig-1*.
- g). Localization of DVE-1 to DD nuclei is essential for synapse elimination.
- h). Analysis of putative DVE-1 target genes and pharmacology implicate the ubiquitin-proteasome system in synapse elimination.

We thank the reviewer for the positive assessment.

Major comments:

Line 369: “DVE-1 transcriptional targets” can change to “putative direct target genes of DVE-1”. Such change is needed because a TF binding event on the genome does not always have an effect on gene transcription. The authors also make great use of available ChIP-Seq data, but more details are needed. What developmental stage was used for DVE-1 ChIP-Seq? If a fosmid-based reporter was used, then the overexpression of DEV-1 could affect, for example, its binding affinity, and this limitation should be acknowledged. Lastly, given that the binding preference (motif) of DVE-1 is not known, can the authors use the coordinates of DVE-1 ChIP-Seq peaks and conduct de novo motif analysis? Even a simple analysis of genomic distribution of the peaks (e.g., promoter, intron, etc) could be informative.

Thanks for pointing out these clarifications. We changed “DVE-1 transcriptional targets” to “putative direct DVE-1 target genes” (line 387 and elsewhere) as suggested by the reviewer. We also used the coordinates of the DVE-1 ChIP-seq data to conduct de novo motif analysis and identified four partially overlapping consensus motifs. Two of these motifs were also recently reported by another group (Shao

et al., 2020), while two are predicted from our work. The ChIPseq studies were conducted on late stage embryos using an integrated DVE-1::GFP translational reporter. We now include this information and acknowledge potential limitations stemming from reporter overexpression (lines 404-406, 402-404).

The data suggest very intriguing non-cell autonomous effects: cholinergic presynaptic sites onto DD neurons persist into adulthood in *dve-1* mutants, and *dve-1* promotes destabilization of vesicle assemblies in cholinergic neurons. First, these appear to be two distinct non-cell autonomous phenotypes, and the authors should clarify this in the text. Second, at a mechanistic level, can the authors discuss what triggers these non-cell autonomous effects (e.g., a secreted factor under DVE-1 transcriptional control)?

The reviewer raises a good point. The cholinergic presynaptic terminals in DA/DB axons form synaptic connections with both muscles and GABA neurons. We speculate that AZ material is preserved because the structural connections with muscles are maintained in the wild type (though we have not tested this directly), while the identity of the GABAergic postsynaptic partner in the dorsal nerve cord changes from DD to VD during the course of wild type remodeling. Our photoconversion experiments provide evidence that wild type cholinergic synaptic vesicle assemblies undergo significant redistribution during this remodeling period, likely due to the removal of juvenile connections onto DD neurons and the formation of new connections with VD neurons. At this point, it remains unclear how presynaptic vesicle assemblies are stabilized in *dve-1* mutants. We can speculate that direct or indirect transcriptional regulation of a retrograde secreted factor or synaptic adhesion process may be involved in a mechanism for destabilization. Though interesting, we hope the reviewer will agree that answering this question lies outside the scope of the present manuscript. We now include discussion of these points (lines 520-523).

The paper states that DVE-1 is implicated in the transcriptional control of the ubiquitin-proteasome system in synapse elimination. Although there is evidence (e.g., GO analysis on DVE-1 targets, proteasome inhibition experiments) supporting this possibility, the authors have yet to formally establish a transcriptional link, i.e., to identify which direct targets of DVE-1 are important for proteasome function and synaptic remodeling. Moreover, the proteasome inhibitor (bortezomib) can have broad effects on degradation of various proteins (because high dosage leads to larval arrest), and this could confound data interpretation. For example, can the authors test whether bortezomid has a specific effect on the iAChR phenotype and/or cholinergic presynaptic sites, or does it also affect

degradation of other proteins that have no role on DD synaptic remodeling? In other words, how “healthy or functional” are the DD neurons after bortezomid treatment?

We appreciate the reviewer’s point here. Please also see response to Reviewer 1 (point 2) and Reviewer 2 (point #8). We used several approaches to strengthen the link between *dve-1* control of remodeling and the ubiquitin-proteasome system. We tested a temperature sensitive allele of the sole E1 homolog, *uba-1*, and found that juvenile AChRs were not properly removed (Figure 7E,F). We also investigated remodeling in several available strains carrying single gene mutations in candidates implicated from the ChIPseq analysis but none of these single gene mutations appreciably impacted remodeling, possibly due to redundancy. Related to the concern that bortezomib treatment may have generalized effects on health, we show that bortezomib-treated animals are able to progress through development and show that dendritic spines, a morphological feature of mature DD neurons, form normally (Figure S7.2G), suggesting that maturation of the GABAergic neurons is not irreversibly disrupted by bortezomib treatment.

Minor comments:

Because DA and DB form dyadic synapses onto GABA and dorsal muscle, the synaptic analysis of *dve-1* can be strengthened by using a marker for ACh receptors (e.g., UNC-38) expressed in dorsal muscle. Along the same lines, the surprising result on the active zone marker UNC-10 can be corroborated with a second marker, such as SYD-1.

We now include additional analysis of ACR-16 ACh receptors in dorsal muscles (Figure S3.1I) as well as an additional cholinergic active zone marker ELKS-1 (Figure 3C, Figure S3.1J).

- In Abstract: “DVE-1 targets” should be “putative DVE-1 target genes”

Agreed. Changed as suggested.

- Figure 1D illustrates that DVE-1 contains two homeodomains with similarity to the HOX homeodomain. In *C. elegans*, only 6 of the ~100 homeodomain transcription factors belong to the HOX family, and DVE-1 is not a HOX gene. Because there is huge confusion in the literature that every homeodomain TF is a Hox gene (which is not true), I would suggest to simply indicate in Fig 1D that DVE-1 has two homeodomains, essentially to change “HOX” to “HD” in the figure, as previous SATB studies have done:

<https://doi.org/10.1093/nar/gkm1151>

<https://pubmed.ncbi.nlm.nih.gov/20351170/#&gid=article-figures&pid=fig-1-uid-0>

Good point. We changed the domain labels from Hox to HD in Figure 1C.

Line 157: was the wild-type *dve-1* gene expressed under a GABA-specific promoter? It is important to state this clearly in text, as the paper argues for a cell-autonomous DVE-1 function.

We apologize this was not more clear. We previously referred to genomic rescue of *dve-1* using native promoter elements and we have clarified this in the text (lines 160-162). In addition, we now show GABA-specific rescue in Figure 1D-E , providing direct evidence for a cell autonomous DVE-1 mechanism of action.

Can the authors rule out the possibility that *ulp-4* affects the transcription of *dve-1*?

Gao et al. 2019 showed downregulation of *ulp-4* affects nuclear localization of wild type DVE-1, but not the engineered DVE-1 K327R variant where the SUMOylation site has been mutated. Gao et al. found that the DVE-1 K327R variant is normally localized to the nucleus in *ulp-4* RNAi-treated animals with no decrease in fluorescence. These findings indicate that ULP-4 regulates DVE-1 SUMOylation rather than expression. Our analysis of DVE-1 K327R in GABA neurons of *ulp-4* mutants is consistent with these findings (Figure 6B,C).

- Line 282: 6G should be 6F.

Fixed.

- The relevance of SATB to human syndromes could be mentioned earlier (e.g., Abstract, Intro).

We now include this point in the introduction.

REVIEWERS' COMMENTS

Reviewer #1 (Remarks to the Author):

The authors have fully satisfied my concerns and addressed all points. The manuscript is much improved and i have no further requests

Reviewer #2 (Remarks to the Author):

The authors have thoughtfully addressed reviewers' comments with plenty of additional data which support their conclusion.

I am in full support of publishing the work in Nature Communications.

Reviewer #3 (Remarks to the Author):

Through additional experiments and text changes, the authors have addressed all my concerns, as well as the issues raised by the other two reviewers. The revised manuscript is now very much improved.

Response to reviewer comments

All 3 reviewers were satisfied with our previous revisions (comments appended below). We appreciate their time and that of the editor and staff. We have reformatted according to journal guidelines as requested.

Reviewer #1 (Remarks to the Author):

The authors have fully satisfied my concerns and addressed all points. The manuscript is much improved and i have no further requests

Reviewer #2 (Remarks to the Author):

The authors have thoughtfully addressed reviewers' comments with plenty of additional data which support their conclusion.

I am in full support of publishing the work in Nature Communications.

Reviewer #3 (Remarks to the Author):

Through additional experiments and text changes, the authors have addressed all my concerns, as well as the issues raised by the other two reviewers. The revised manuscript is now very much improved.